# Observed and projected global warming pressure on coastal hypoxia

Michael M. Whitney[1]

[1]Department of Marine Sciences, University of Connecticut, 1080 Shennecossett Road, Groton, CT, USA

*Correspondence to*: Michael M. Whitney (michael.whitney@uconn.edu)

5   **Abstract.** Coastal hypoxia is a major environmental problem of increasing severity. A global 40-year observational gridded climate data record and 21[st] century projections from the Community Earth System Model (CESM) under RCP8.5 forcing are analyzed for long-term linear trends in summer-month conditions, with a focus on warming-related pressures on coastal oxygen levels. Projected surface temperature and oxygen conditions are compared to global observations over the 16-year overlapping period (2006-2021). Median linear trends for 2006-2100 along the global coast are 0.32 °C, -1.6 mmol m$^{-3}$, and -1.2 mmol m$^{-3}$ per decade for sea-surface temperature (SST), oxygen saturation concentration at the surface (surface oxygen capacity), and vertical-minimum oxygen concentration, respectively. These trends point to more rapid deterioration in coastal conditions than experienced over recent decades; the projected median coastal trends for SST and oxygen capacity are 148% and 118% of the corresponding observed rates. Companion analysis of other models and climate scenarios indicates projected coastal oxygen trends for the more moderate RCP4.5 and updated SSP5-8.5 scenarios respectively are 37-77% and 103-196% of the CESM RCP8.5 projections. Median rates for the coast and documented hypoxic areas are higher than in the global ocean. Warming and oxygen declines tend to be fastest at high latitudes, one region where new hypoxic areas may emerge as oxygen conditions deteriorate. There is considerable pressure on current hypoxic areas since future oxygen declines of any magnitude will make hypoxia more severe. The projections can inform coastal environmental management strategies to protect future water quality and ecosystem services.

## 1 Introduction

Hypoxia in coastal waters is a major environmental problem of increasing severity confronted around the world (Hoegh-Guldberg et al., 2018). "Dead zones in the coastal oceans have spread exponentially since the 1960s and have serious consequences for ecosystem functioning" (Diaz and Rosenberg, 2008). To date, over 500 hypoxic areas in estuaries, coastal seas, and on continental shelves have been documented globally (Breitburg et al., 2018). Furthermore, the severity of hypoxia has increased in many areas (Rabalais et al., 2010). Coastal oxygen concentrations (within 30 km from the global coast) have been decreasing an order of magnitude faster than surface-layer concentrations in the open ocean (Gilbert et al., 2010). Worsening coastal oxygen conditions are attributable to the dual human pressures of nutrient overloading fueling eutrophication and anthropogenic climate change (e.g. Rabalais and Turner, 2001; Paerl, 2006; Diaz and Rosenberg, 2011). Coastal oxygen conditions are influenced by many aspects of climate controls including warming waters, altered storm patterns, changing precipitation and river flow, sea-level rise, and shifting ocean circulation (Altieri and Gedan, 2015). This

paper focuses on warming-related pressures on coastal hypoxia. Oxygen saturation concentration (oxygen capacity) decreases with increased water temperatures (Weiss, 1970; Garcia and Gordon, 1992), which can exacerbate hypoxia. Oxygen capacity is a succinct synonym for oxygen saturation concentration which is borrowed from physiological research on blood oxygen levels (e.g. Haldane and Smith, 1900; Black, 1940; Maio and Neville, 1965; Bernal et al., 2018) and has been applied to

dissolved oxygen in the coastal and open ocean (Helm et al., 2011; Deignan-Schmidt and Whitney, 2017). Metabolic rates and related oxygen demands also rise with temperature (Brown et al., 2004). Warming also can intensify and extend the duration of summertime thermal stratification that inhibits ventilation of near-bottom hypoxic waters (Cloern, 2001).

Earth system models provide projections of 21$^{st}$ century conditions relevant to coastal hypoxia. Altieri and Gedan (2015) analyzed sea surface temperature (SST) projections from the Community Climate System Model (Collins et al., 2006) for the

greenhouse gas emissions A1B scenario (Nakicenovic and Swart, 2000). Under this emission scenario, almost all documented coastal hypoxic areas (based on the Diaz and Rosenberg (2008) dataset) would experience 2 $^{o}$C or greater water temperature rise by the century end (Altieri and Gedan, 2015). There is a need to update this type of analysis with more recent climate modeling and analyze oxygen trends for Earth system models that incorporate ocean ecosystem dynamics. The present study analyzes Community Earth System Model (CESM) results that include ocean biogeochemistry (Kay et al., 2015a) for the

Representative Concentration Pathway (RCP) 8.5 (Moss et al., 2010); which is part of the Climate Model Intercomparison Project phase 5 (CMIP5). The RCP8.5 exhibits more global warming than the earlier A1B emission scenario (Melillo et al., 2014). The study also compares CESM RCP8.5 results to CMIP5 and CMIP6 for different models and climate scenarios.

Earth system models offer great value in projecting future conditions, but they do have limitations that affect coastal applications. The relatively course horizontal and bathymetric resolution of CESM and most Earth system models, particularly

those including ocean biogeochemistry, limits the representation of coastal processes. Earth system models do not currently resolve estuary ecosystems, but they do have the resolution to represent coastal conditions at regional scales. Regional patterns strongly influence rates of change for coastal SST (e.g. Pershing et al., 2015) and temperature-related changes in oxygen capacity. Coastal oxygen concentrations should vary with the regional variations in temperature and oxygen capacity resolved by Earth system models, but oxygen levels also depend on other abiotic and biotic factors. Oxygen concentrations tend to be

highly variable on small spatial scales that are not resolved by CESM, particularly within smaller estuarine waters. For these reasons, projected changes for coastal oxygen concentrations may be less reliable than SST and oxygen capacity. Open-ocean results at 300 m (Oschlies et al., 2017) and 100-600 m (Cocco et al., 2013) point to model limitations in representing the observed distribution of dissolved oxygen trends. It is reasonable to expect model-observation oxygen mismatches also occur along coasts. The projections for temperature and oxygen changes, nevertheless, are worth considering and evaluating given

the importance of anticipating potentially worsening conditions in coastal hypoxic areas. Projections from Earth system models should be considered in the context of observed trends in coastal conditions and compared to available global coastal observations. Gridded SST climate data records can provide sufficient global coastal data coverage and allow for computation of oxygen capacities to evaluate whether Earth system models provide reasonable representations of coastal conditions affecting hypoxia. Comparison to estuarine oxygen conditions can be made using long-term observations within some

estuaries, but is not practical for all estuaries globally. Frankly discussing the limitations and appropriate application of Earth system model projections for hypoxic estuaries is important and should motivate future modeling improvements.

The main objective of this paper is to quantify global patterns exacerbating coastal hypoxia by analyzing linear trends in SST, surface oxygen capacity, and vertical-minimum oxygen concentrations (the minimum dissolved oxygen in the water column at each location). Observations from a satellite-derived SST global climate data record (Merchant et al., 2019; Embury and

Good, 2021) are analyzed for coastal SST and oxygen-capacity trends over the last four decades, which provide context for the projections. New analysis of 21$^{st}$ century RCP8.5 projections from the CESM Large Ensemble Project (Kay et al., 2015a,b) is completed for coastal areas and compared to open-ocean rates. Observed and projected coastal SST and oxygen capacities are compared for the first 16 years of the projection period that already have occurred. The study not only investigates projections for documented coastal hypoxic locations, but also considers the entire global coast to include unknown and

potentially emerging hypoxic areas. Companion analysis of other CMIP5 RCP8.5 models, the more moderate RCP4.5 scenario, and the corresponding updated CMIP6 Shared Socioeconomic Pathways (SSP) 2-4.5 and 5-8.5 provides context for the CESM RCP8.5 coastal results. Comparisons to prior studies are included and applications to some well-studied hypoxic estuaries are discussed. Implications for future coastal hypoxia and management are discussed, limitations of current Earth system model projections are considered, and modeling techniques for better projections of coastal oxygen conditions are recommended.

**2 Methods**

**2.1 Observations**

The global observational dataset analyzed is the satellite-based SST time-series described in Merchant et al. (2019) and available with updates at the Climate Data Store of the Copernicus Climate Change Service (Embury and Good, 2021). The Level-4 (version 2.0) product combines SST data from several satellite platforms to construct a high-quality climate data

record that has been validated with *in situ* observations. The dataset is a daily product on a regular grid with 0.05$^{o}$ (latitude and longitude) resolution. The Level-4 product is gap-filled so that each grid point has an SST value for every day from September 1981 up to within a month of present time. The SST data variable used in this study has been corrected from skin temperature to a nominal depth of 0.2 m in order to pair well with *in situ* surface observations (Merchant et al., 2019). This study analyzes 40 years of data spanning 1982-2021. Daily data during each August and February are used to represent summer conditions in

the northern and southern hemispheres, respectively. The rationale for analyzing these months is they are the summer months in each hemisphere when water temperatures tend to be highest and oxygen levels tend to be lower. Daily data are averaged to create August-averaged and February-averaged SST time series for the northern and southern hemispheres, respectively. Oxygen capacities are the oxygen saturation concentrations calculated with the Garcia and Gordon (1992) equations using the monthly averaged SST data and a constant 35 salinity. The constant salinity is used because the Merchant et al. (2019) product

does not include salinity and because this straightforward approach is sufficient to provide observational context for the projections. The sensitivity of oxygen capacity trends to the choice of salinity is assessed with different salinities specified in

the calculations. Coastal points are defined as grid points with at least one neighboring land cell (directly to the east, west, north, or south).

## 2.2 CESM RCP8.5 projections

Projections of 21[st] century water temperatures and oxygen conditions are derived from the CESM Large Ensemble Project that includes ocean biogeochemistry (Kay et al., 2015a), which is a contributor to CMIP5 (Taylor et al., 2012). Monthly-averaged results for 2006-2100 are accessed via the Earth System Grid (Kay et al., 2015b). Multiple ensemble members for RCP8.5 forcing (following CMIP5 protocols) are used: 35 and 29 ensemble members for temperature and oxygen variables, respectively. Runs 1-35 are included for SST (the "SST" variable) and temperature at fixed vertical levels (the "TEMP"

variable). Runs 1, 2, and 9-35 are analyzed for surface oxygen capacity (saturation concentration at the surface, the "O2SAT" variable), vertical-minimum oxygen concentration (minimum dissolved oxygen concentration in each water column, the "O2_ZMIN" variable), oxygen concentrations at fixed vertical levels (the "O2" variable), and the apparent oxygen utilization at the same levels (the "AOU" variable). Runs 3-8 are omitted for oxygen variables because these results are not available on the Earth System Grid. Additional runs (101 and higher) were avoided because of a documented unexplained positive bias in

global temperature relative to other ensemble members. The surface level (representing 0-10 m depth) is subsampled from the O2 variable to obtain surface oxygen concentrations. All included runs are averaged together to produce ensemble-mean monthly time series of SST, surface oxygen capacity, surface oxygen concentration, and vertical-minimum oxygen concentration at each ocean grid cell. Results from individual runs are analyzed to assess uncertainty associated with differences among ensemble members. Supplementary analysis of temperatures, oxygen concentrations, and AOU at 10-m

intervals down to 100 m deep is included to describe transitions from surface conditions to deeper levels in the coastal water column.

For each year at each CESM ocean grid cell (at 1° latitude nominal resolution), the month with minimum surface oxygen capacity is selected to construct annual time series of minimum oxygen capacity and coincident SST and oxygen concentrations. August and February are the median months associated with minimum surface oxygen capacity in the northern

and southern hemispheres, respectively. The global set of all coastal points includes all CESM ocean grid cells with at least one neighboring land cell (there are 4,899 coastal cells).

## 2.3 Comparison of observations and projections

To characterize the reliability of CESM in coastal areas, observations and projections are compared for global coastal points. There are 16 projection-period years spanning 2006-2021 that now overlap with observed conditions. Every CESM coastal

point is matched with the closest coastal point on the observational grid, which has higher spatial resolution than the CESM grid. The summer-month values (calculated as described in previous section) for the overlapping period are averaged together to determine mean observed and projected values at each coastal point. The resulting mean SST and oxygen capacity values are used to assess local and global biases relative to observations. Linear regression is completed to indicate the global

relationship between observed and projected SST and oxygen capacity for coastal waters. Regression results are included in the supporting dataset (Whitney, 2021) and are reported with the slope, offset, p-value (p) of the F-statistic, correlation coefficient squared ($r^2$), root mean square error (RMSE), and standard error of the regression slope ($\sigma_s$). The level of agreement between observed and projected mean values for the overlapping period assesses the quality of CESM performance in coastal waters. Observed and projected temporal trends are not compared for the overlapping period because observations have high interannual variability that can obscure longer-term trends over shorter periods. Individual ensemble members have comparably large interannual variations, but the ensemble-mean results have much less interannual variability.

## 2.4 Trend analysis

Linear regression analysis is applied to characterize long-term temporal trends. It is worth noting that nonlinear trends are possible, but the strategy is to focus on constant long-term rates of change and the results indicate linear trends fit the data well in most locations globally. For the observations, the entire 40-year observational period is regressed with time for SST and surface oxygen capacity. For the CESM RCP8.5 projections, the entire 94-year projection period is regressed for ensemble-mean SST, surface oxygen capacity, and surface and vertical-minimum oxygen concentrations. The regression slopes are reported as rates of change associated with each variable and are included in the supporting dataset (Whitney, 2021). These rates are graphed spatially and the spatial median rates are reported for the global ocean, coastal grid cells, and documented hypoxic areas. Regression statistics are calculated. The p-values of the F-statistic are used to categorize results depending on whether or not they are statistically significant at the 90% confidence level (p=0.10). Correlation coefficients are calculated but not emphasized because ensemble-mean projections average out interannual variability present in ensemble members and therefore tend to have high $r^2$ values; while the observational record has higher interannual variability and correspondingly lower $r^2$ values. Uncertainty associated with the regression fit is quantified with $\sigma_s$, the standard error of the regression slope. Based on the $\sigma_s$ values, the corresponding standard error of the spatial median values ($\sigma_m$) are reported for global, coastal, and documented hypoxic areas. For CESM RCP8.5 projections, variability among ensemble members is characterized by calculating regression slopes for each member and then calculating the associated standard error of the ensemble-mean rates of change ($\sigma_e$) at each grid cell. The spatial median rates for each ensemble member also are calculated and the corresponding standard deviation of these median rates ($\sigma_{me}$) are reported as another way to characterize intra-ensemble variability. Rates of change are shown for the entire global ocean and emphasis is placed on change in coastal waters.

## 2.5 Documented coastal hypoxic areas

The 532 documented coastal hypoxic areas analyzed are from the Diaz et al. (2011) database. The dataset is an expanded version of the Diaz and Rosenberg (2008) database, which is similar to the coastal dataset in Breitburg et al. (2018). The oxygen concentration threshold used to identify hypoxia in the dataset is 63 mmol m$^{-3}$ (2 mg L$^{-1}$) (Diaz and Rosenberg, 2008). The Diaz et al. (2011) database also includes 244 additional locations classified as eutrophic (but not hypoxic); these documented eutrophic points are not isolated as a group in this analysis, but are included among the global coastal points

analyzed. Rates of change (for temperature and oxygen variables) for the documented hypoxic areas are extracted from rates at the nearest observational coastal grid point or CESM coastal ocean grid cell. The observational grid resolution is fine enough to represent each individual documented coastal hypoxic area with a separate observational grid point. The coarser (nominally 1 ° latitude) CESM resolution causes some documented locations to be represented by the same CESM coastal cell. The 532 documented locations map to only 240 unique CESM ocean grid cells.

## 2.6 Other projections

Projections from other models and climate scenarios provide context for the CESM RCP8.5 ensemble results. Data are accessed online via the Earth System Grid Federation. Only models with ocean biogeochemistry are included. Monthly-averaged results for vertical-minimum oxygen concentrations (the "o2min" variable) are analyzed. The CMIP5 RCP4.5 and RCP8.5 sets include five other models: HadGEM2-ES, IPSL-CM5A-LR, IPSL-CM5A-MR, MPI-ESM-LR, and MPI-ESM-MR (Collins et al., 2011; Dufresne et al., 2013; Giorgetta et al., 2013). The CESM2 (CMIP6) SSP2-4.5 and SSP5-8.5 sets include the three available ensemble members (r4i1p1f1, r10i1p1f1, and r11i1p1f1) with oxygen data (Danabasoglu et al., 2020). The CMIP6 SSP5-8.5 set of other models includes CanESM5 (r1i1p2f1 run), IPSL-CM6A-LR, MPI-ESM1-2-LR, and MPI-ESM1-2-HR (Swart et al., 2019; Boucher et al., 2020; Mauritsen et al., 2019). The r1i1p1f1 runs are selected, unless already noted. Oxygen trends are calculated following the exact same procedures as described above. Analysis years are 2006-2100 and 2015-2100 for CMIP5 and CMIP6 models, respectively.

## 3 Results

### 3.1 Observed trends

The 40-year observational SST record (updated from Merchant et al., 2019) indicates warming has occurred throughout the world ocean (Fig. 1a). SST rates are stronger in the Northern Hemisphere, with the most rapid warming occurring in Arctic areas near coasts. Just over half (55%) of the ocean grid points have linear trends with $p \leq 0.10$ (Table 1). P-values are higher where calculated rates are lower in parts of the Atlantic, much of the South Pacific, and most of the Southern Ocean. Where $p \leq 0.10$, the mean $r^2 = 0.20$; indicating that linear trends describe an appreciable part of the observed variance, but interannual variability is larger. Since the long-term linear trends describe only part of the variance, the uncertainty of the regression slopes is relatively large. Among the locations where $p \leq 0.10$, the average $\sigma_s$ is 30% of the local rate of change. The observed global median SST trend (including only points with $p \leq 0.10$) is 0.22 °C per decade (Table 1). The $\sigma_m$, the standard error of the global median trend, is exceedingly small ($2 \times 10^{-4}$ °C per decade) because of the large number of data points (Table 1). The median

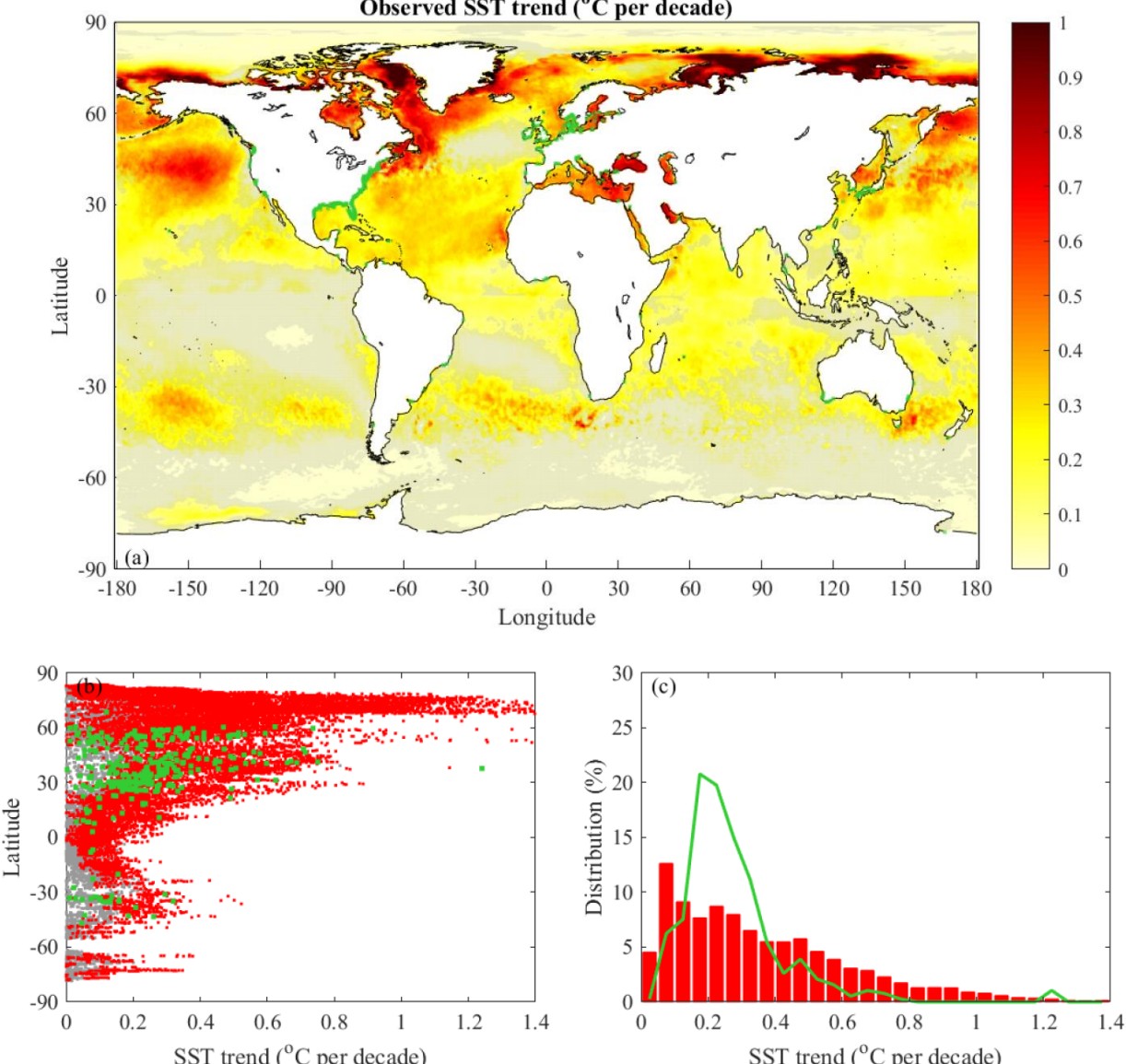

**Figure 1: Observed SST trends: (a) spatial distribution for all ocean observational points, (b) latitudinal dependence for global coastal points and documented coastal hypoxic areas, and (c) histograms of global coastal points and documented coastal hypoxic areas (having linear trends with p≤0.10). Locations that have linear trends with lower confidence (p>0.10) are marked with grey points. Global coastal data are shown as red points and bars. Documented coastal hypoxic areas are marked in green.**

trend with all points included (regardless of p-value) is 0.13 °C per decade. This rate is consistent with the global mean SST rate of 0.1°C per decade from 1982-2013 (Pershing et al., 2015).

Observed rates along the global coast indicate conditions relevant to coastal hypoxia. The median SST trend for global coastal points (0.27 °C per decade, for points with p≤0.10) is 22% faster than the corresponding ocean median rate (Table 1). SST

rates tend to be near the median coastal value from 60 °S to 30 °N (Fig. 1b) and increase towards higher latitudes. Similar to the global points, the average $\sigma_s$ is 34% of the local rate of change for coastal points with $p \leq 0.10$. The $\sigma_m$ is much smaller than the median coastal value (Table 1). The observed median trend for documented hypoxic areas (0.24 °C per decade) is 6% faster than the ocean median rate (Table 1). The rate in documented hypoxic areas is somewhat (11%) lower than the median global coastal rate because there are few documented hypoxic areas at high latitudes where warming is fastest. The histograms of SST rates (Fig. 1c) indicate the documented hypoxic areas have a narrower spectrum than the global coastal points.

The global distribution of oxygen capacity trends (Fig. 2a) at the surface is tied to observed SST rates (Fig. 1a). Since oxygen capacity decreases nonlinearly with temperature with a greater response at lower temperatures, oxygen capacity decreases are amplified at high latitudes where waters are colder and projected warming rates are faster (Weiss, 1970; Garcia and Gordon, 1992; Altieri and Gedan, 2015). Oxygen capacity linear trends with $p \leq 0.10$ occur in the same locations as for SST trends and $r^2$ values are similar to those for SST. The average $\sigma_s$ is 38% of the local rate of change. The observed global median oxygen capacity trend at the surface (including only points with $p \leq 0.10$) is -0.8 mmol m$^{-3}$ per decade and $\sigma_m$ is very small (Table 1). The calculated global median rate is several times faster than the median rate of -0.2 mmol m$^{-3}$ per decade observed in offshore (>100 km from the coast) upper-ocean (0-300 m) waters for 1976-2000 (Gilbert et al., 2010). The mismatch with the prior study is likely due to the more recent time period and the reliance on surface, rather than upper-ocean, observations. Due to these methodological differences, matching rates between studies is not expected, but the earlier study does provide context.

**Table 1: Observed (1982-2021) and projected (2006-2100) median linear trends in SST, surface oxygen capacity, surface oxygen concentrations (projected only), and vertical-minimum oxygen concentrations (projected only) for the ocean, global coastal points, and documented coastal hypoxic areas. All points included in median calculations have linear trends with $p \leq 0.10$, the corresponding percent coverage of $p \leq 0.10$ is given in square brackets. Median rates are listed with $\pm\sigma_m$, the standard errors of the median values.**

| Rate | Ocean | Coastal | Hypoxic |
|---|---|---|---|
| Observed SST (°C per decade) | 0.218 ±2x10$^{-4}$ [55.12%] | 0.265 ±10$^{-3}$ [70.64%] | 0.236 ±0.01 [76.13%] |
| Observed oxygen capacity (mmol m$^{-3}$ per decade) | -0.82 ±10$^{-3}$ [55.06%] | -1.34 ±0.01 [70.64%] | -0.78 ±0.04 [76.13%] |
| Projected SST (°C per decade) | 0.352 ±3x10$^{-5}$ [100%] | 0.392 ±2x10$^{-4}$ [100%] | 0.402 ±2x10$^{-4}$ [100%] |
| Projected oxygen capacity (mmol m$^{-3}$ per decade) | -1.23 ±2x10$^{-4}$ [100%] | -1.58 ±2x10$^{-3}$ [99.98%] | -1.39 ±10$^{-3}$ [100%] |
| Projected surf. oxygen conc. (mmol m$^{-3}$ per decade) | -1.33 ±6x10$^{-4}$ [99.19%] | -1.51 ±4x10$^{-3}$ [98.12%] | -1.41 ±2x10$^{-3}$ [100%] |
| Projected vert.-min. oxygen conc. (mmol m$^{-3}$ per decade) | -0.65 ±4x10$^{-4}$ [97.24%] | -1.15 ±3x10$^{-3}$ [97.39%] | -1.38 ±5x10$^{-3}$ [100%] |

The observed median oxygen capacity trend for global coastal points (-1.3 mmol m$^{-3}$ per decade, for points with $p \leq 0.10$) is 62% faster than the surface ocean median rate (Table 1). Similar to the global points, the average $\sigma_s$ is 34% of the local rate of change for coastal points with $p \leq 0.10$ and $\sigma_m$ is much smaller than the median coastal value (Table 1). The observed median

coastal rate is half of the rate calculated for a global coastal band (within 30 km of the coast) for 1976-2000 (Gilbert et al., 2010). For the reasons mentioned above, a match between the studies is not expected. It is interesting that the current study has faster global rates and slower coastal rates than the Gilbert et al. (2010) study; the underlying reasons are not explored here. Rates tend to be near the median coastal value from 60 ºS to 30 ºN (Fig. 2b). At higher latitudes, rates tend to increase with latitude. The median oxygen capacity trend for documented hypoxic areas is -0.8 mmol m$^{-3}$ per decade (Table 1). This rate is lower magnitude than for all global coastal points because of little high-latitude coverage, where oxygen capacity rates are largest. Similar to SST rates, the histograms for oxygen capacity (Fig. 2c) indicate the documented hypoxic areas have a narrower spectrum than the global coastal points. Overall, the observational SST and oxygen capacity analysis provides context for the projections and new information about global coastal conditions influencing coastal hypoxia.

As described in the methods, oxygen capacity is calculated from SST and a constant 35 salinity. This approach neglects changes in oxygen capacity due to long-term salinity variability and the constant salinity choice may bias the calculated oxygen capacity trends. Sensitivity calculations with constant salinity values of 32 and 34 had RMSE (relative to the original calculations) of at most 10$^{-3}$ mmol m$^{-3}$ per decade. Introducing a long-term linear salinity trend of 34.9 to 35.1 (or vice versa) over the observation period created similarly small RMSE of 3x10$^{-3}$ mmol m$^{-3}$ per decade. The sensitivity calculations indicate that assuming a constant salinity does not introduce much error in the oxygen capacity trends.

### 3.2 Comparison of observations and projections

The observational SST record is compared to CESM RCP8.5 projected coastal conditions for the overlapping 16 years spanning 2006-2021 (Fig. 3a). As described in the methods, the comparison involves mean summer-month values for the overlapping period rather than comparing time series with relatively short-term interannual variability. There is a nearly one-to-one relationship between observed and projected coastal SST with a low p-value (p<0.001) and high correlation (r$^2$=0.97). The projected temperatures have a small positive bias (0.3 ºC) and a moderate RMSE (1.9 ºC) relative to observations. This level of agreement indicates that CESM results are broadly representative of global coastal SST conditions. The comparison of observed and projected coastal oxygen capacities averaged over the overlapping period (Fig. 3b) also shows a near one-to-one relationship with a low p-value (p<0.001) and high correlation (r$^2$=0.94). Projected oxygen capacities have a bias and RMSE of -5 and 17 mmol m$^{-3}$, respectively. The scatter away from the regression line is larger at higher oxygen capacities (in colder waters). The nonlinear relationship between temperature and oxygen capacity means that temperature scatter in colder waters translates into more oxygen capacity scatter. This analysis points to the degree of reliability of CESM results for representing coastal oxygen capacities. Overall, the comparison of observations and CESM results suggest that CESM can provide reliable projections of coastal conditions relevant to coastal hypoxia.

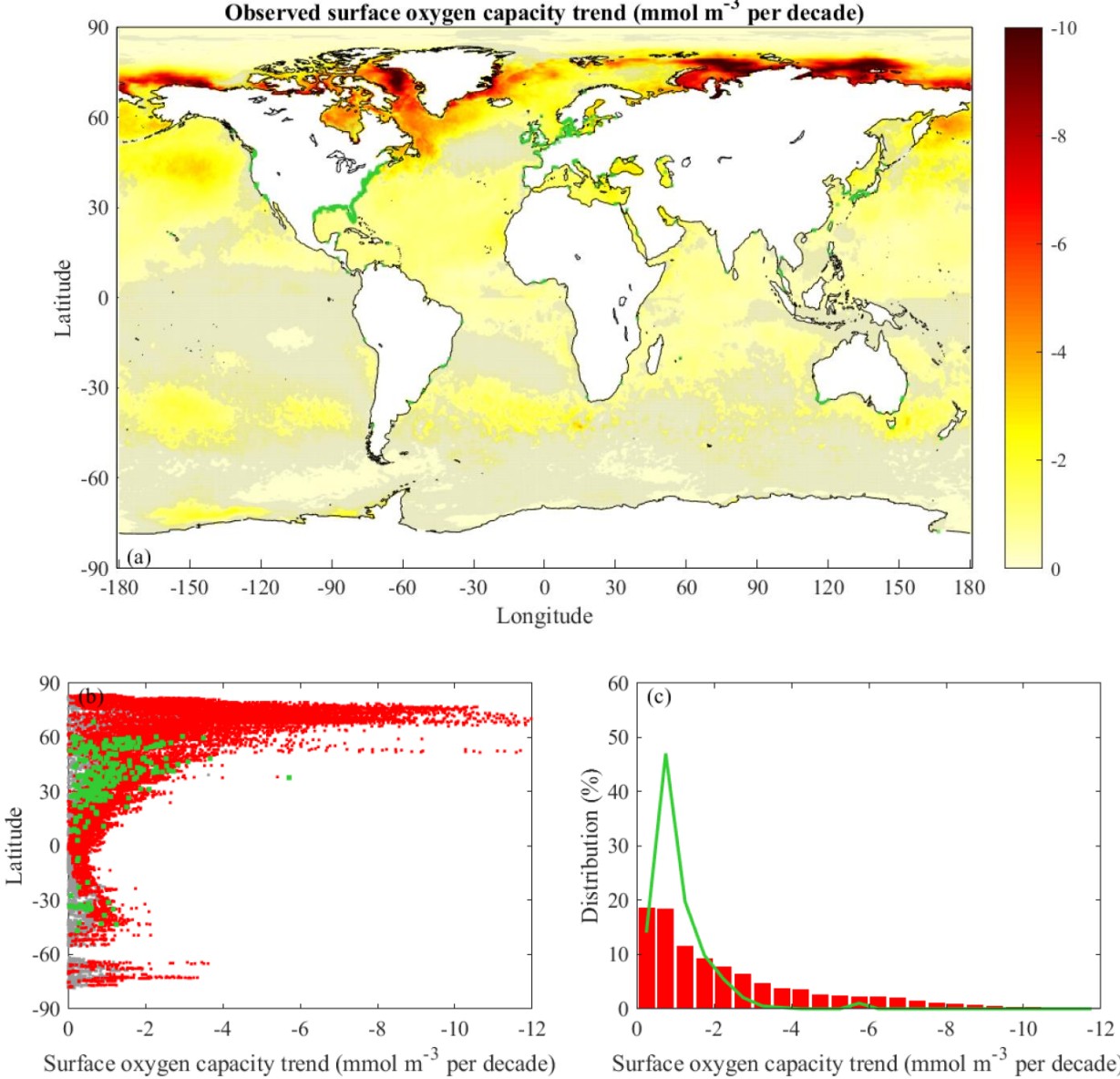

**Figure 2: Observed surface oxygen capacity (saturation concentration) trends: (a) spatial distribution for all ocean observational points, (b) latitudinal dependence for global coastal points and documented coastal hypoxic areas, and (c) histograms of global coastal points and documented coastal hypoxic areas. The format and color coding follow Fig. 1.**

### 3.3 Projected trends

The CESM Large Ensemble Project ensemble-mean projection for the RCP8.5 scenario indicates SST will appreciably increase throughout the world ocean over the 21st century (Fig. 4a). All of the ocean cells have linear trends with p≤0.10 (Table 1). SST linear trends account for more than 90% of the variance in the ensemble-mean projection (r²>0.90) for most of the ocean; the

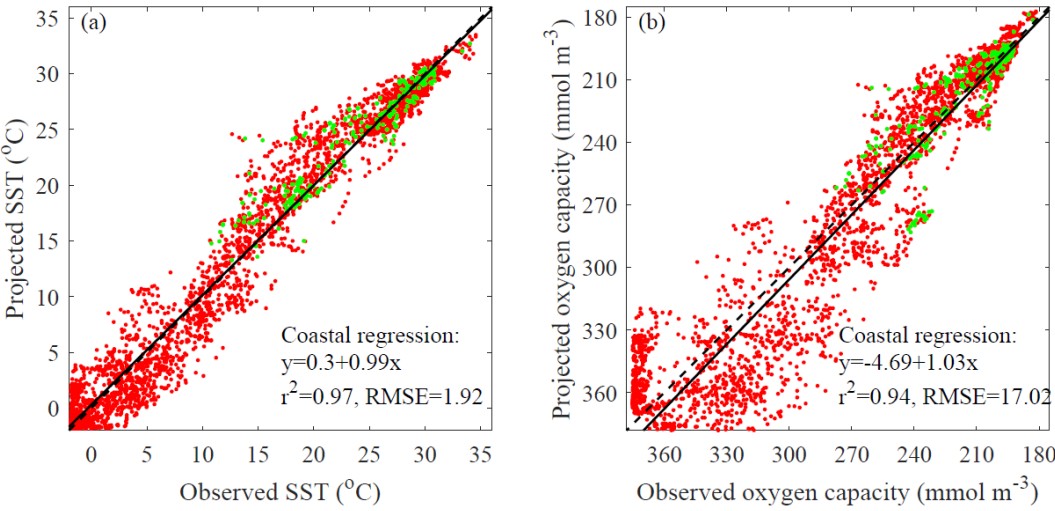

**Figure 3: Comparison of projected to observed coastal conditions: (a) SST and (b) surface oxygen capacity. Mean values for the overlapping 2006-2021 period are compared. Global coastal points and documented coastal hypoxic areas are marked in red and green, respectively. The one-to-one line (dashed) and linear regression (solid) are included along with associated regression statistics (p<0.001 for both regressions).**

only exceptions are some areas to the north and south of Greenland and near parts of Antarctica. The average $\sigma_s$ is only 2% of the local rate of change for points with $p \leq 0.10$. The $\sigma_s$ values for the projections are much smaller than for observations. The projected global median SST trend is 0.35 ºC per decade and the associated $\sigma_m$ is negligible (Table 1). Global distributions of SST warming have been studied in detail for multiple models and RCP scenarios (e.g. Bopp et al., 2013). Bopp et al. (2013) includes CESM simulations in an analysis of ten models running the RCP8.5 scenario and finds the global average SST increase is 0.27 ºC per decade (from the 1990s to 2090s) when averaged across all included models. The smaller warming rate for the Bopp et al. (2013) results is likely connected to including multiple models and the different time period analyzed. The CESM projected global median SST trend is 61% higher than the observed global warming rate (0.22 ºC per decade) calculated in the previous section (Table 1). Under the RCP8.5 scenario, the ocean SST will increase considerably faster than the observed linear trend over the last four decades.

The global distribution of SST warming indicates variations among oceans, with the Arctic Ocean projected to experience at least twice the median warming rates (Fig. 4a). Observations (Fig. 1a) also indicate rapid SST increases near Arctic coasts. There are, however, clear differences in the spatial structure of projected and observed rates. Differences away from the coast in the Arctic Ocean are immediately apparent: the projected SST rates are much stronger than observed rates away from the coasts. These offshore differences are not explored further here, as the focus is on coastal conditions. The high p-values of

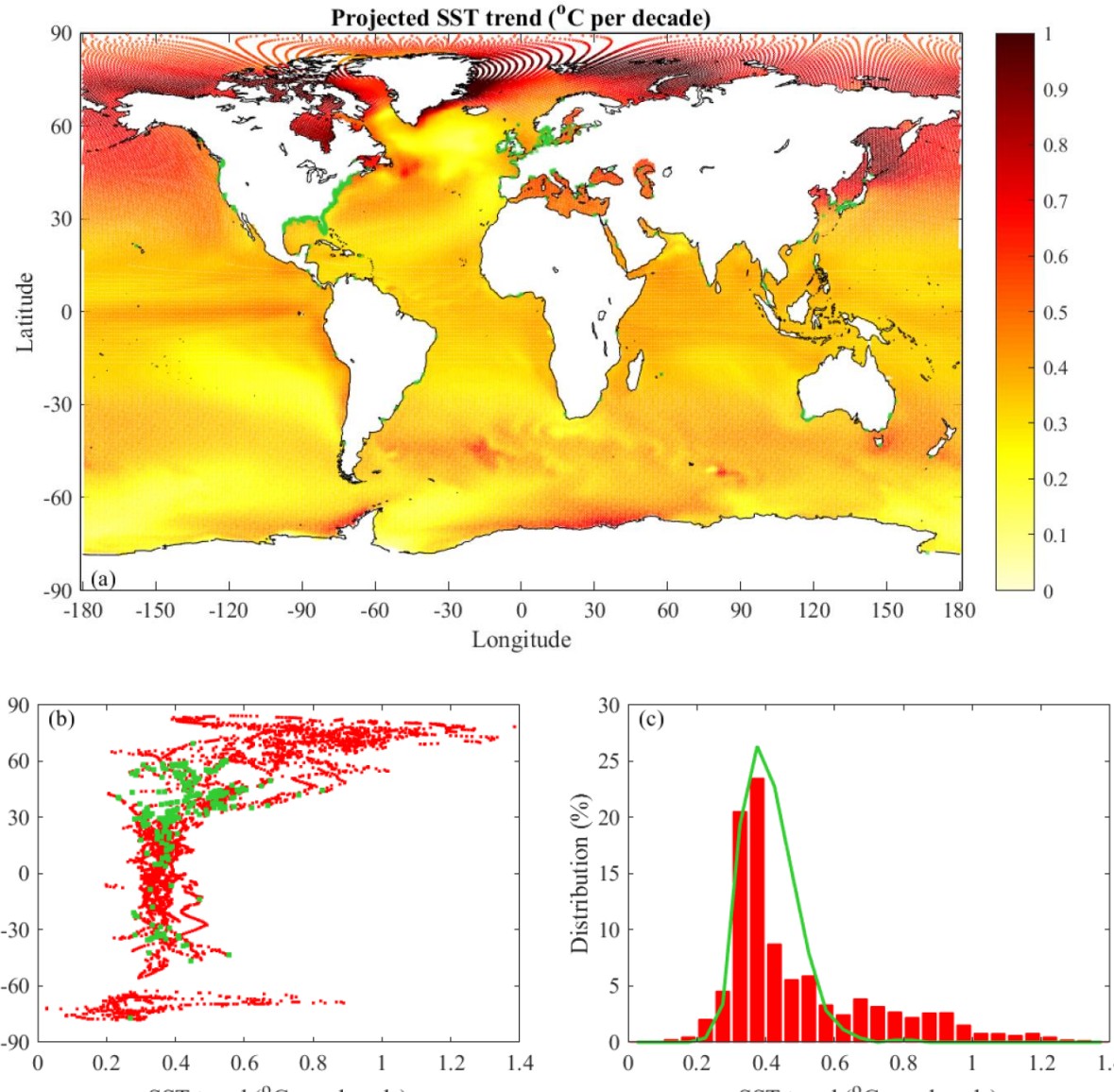

**Figure 4: Projected SST trends: (a) spatial distribution for all CESM ocean grid cells, (b) latitudinal dependence for global coastal points and documented coastal hypoxic areas, and (c) histograms of global coastal points and documented coastal hypoxic areas. The format and color coding follow Fig. 1.**

observed SST trends in much of the Southern Ocean (p>0.10) preclude comparisons of projected and observed spatial structure in this region.

Focusing on warming in global coastal areas reveals new information directly relevant to coastal hypoxia. The projected median SST trend for global coastal points (0.39 ºC per decade) is 11% faster than the projected ocean median rate and 48% higher than the observed median coastal rate (Table 1). Local coastal SST rates (with $\sigma_s$ averaging 2% of the coastal rates)

tend to be near the median coastal value from 60 °S to 30 °N (Fig. 4b). Above 30 °N, warming rates tend to increase with latitude and variability among coasts increases. The latitudinal patterns are broadly similar for projected and observed coastal SST rates (Fig. 4b and Fig. 1b). The projected median SST trend for documented hypoxic areas (0.40 °C per decade) is 14% faster than the projected ocean median rate (Table 1). For documented hypoxic areas, the projected median trend is 70% higher than the median observed rate (Table 1) and 75% faster than the median trend under moderate A1B emissions scenario (Altieri

and Gedan, 2015). The documented hypoxic areas sample much of the variability in coastal SST rates from 45 °S to 60 °N. The hypoxic area database, however, has little coverage at higher latitudes, where the most rapid warming is projected. Histograms (Fig. 4c) indicate most coastal locations (92%) and documented hypoxic areas (96%) are projected to warm more than 0.3 °C per decade; 0.1 °C per decade faster than the A1B results in Altieri and Gedan (2015). A significant portion of the global coast (18%) are projected to warm faster than 0.60 °C per decade, but few of the documented hypoxic sites (2%) are in

this range because most are not at high latitudes. Supplementary trend analysis of temperatures at 10-m depth intervals down to 100 m deep also indicates robust warming trends (not shown), but the coastal warming rates at the 20-30 m, 40-50 m, and 90-100 m levels decrease to 92%, 80%, and 70% the SST trend magnitude, respectively.

The global distribution of projected oxygen capacity trends (Fig. 5a) at the surface is tightly linked to SST rates (Fig. 4a). All of the ocean cells have linear trends with $p \leq 0.10$ (Table 1). The trends have $r^2 > 0.90$ over most of the ocean with exceptions in

the areas where SST rates have $r^2 \leq 0.90$. The $\sigma_s$ values on average are 2% of local rates. The projected global median oxygen capacity trend at the surface is -1.2 mmol m$^{-3}$ per decade and $\sigma_m$ is diminishingly small (Table 1). The median rate is 50% higher than the observed global median trend (Table 1). The projected median rate for ocean waters above 60 °N (-5.3 mmol m$^{-3}$ per decade) is several times higher than the total ocean median rate. Projected oxygen capacity trends in the Arctic Ocean, particularly offshore, are much stronger than observed rates; this Arctic pattern echoes the differences in SST rates.

The projected median oxygen capacity trend for global coastal points (-1.6 mmol m$^{-3}$ per decade) is 28% faster than the surface ocean median rate (Table 1). The projected median coastal rate is 18% faster than observed. Rates tend to be near the median value from 30 °S to 30 °N (Fig. 5b). Outside that latitude range, rates tend to increase with latitude and variability among coasts increases, particularly in the northern hemisphere where rates can exceed -10.0 mmol m$^{-3}$ per decade. The latitudinal pattern in coastal oxygen capacity trends (Fig. 5b) is similar to the observed coastal pattern (Fig. 2b). The median oxygen capacity

trend for documented hypoxic areas is -1.4 mmol m$^{-3}$ per decade (Table 1). This rate is lower magnitude than for all global coastal points because of little high-latitude coverage, where oxygen capacity rates are largest. The projected median trend for documented hypoxic areas is 78% higher than observed (Table 1). Histograms (Fig. 5c) indicate for most coastal locations (95%) and all documented hypoxic areas the oxygen capacity trend is projected to be faster than -0.9 mmol m$^{-3}$ per decade. A significant portion of the global coast (28%) has projected rates faster than -3.0 mmol m$^{-3}$ per decade, but few of the

documented hypoxic sites (4%) are in this range because most are not at high latitudes.

The distribution of surface oxygen concentration trends (not shown) is very similar in terms of magnitudes and spatial patterns to the distribution for oxygen capacity trends (Figure 5). The median surface oxygen concentration trends for global, coastal,

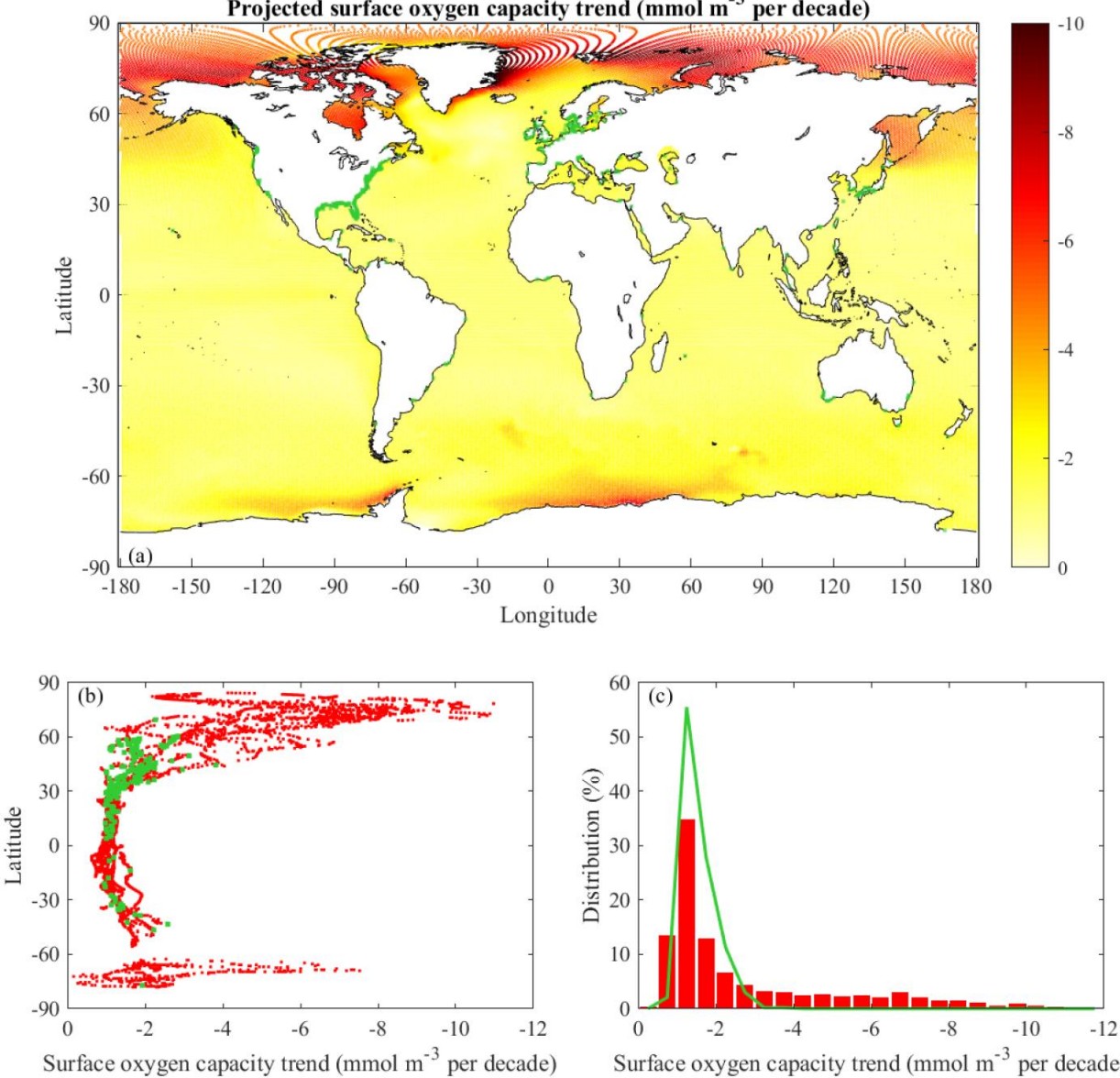

**Figure 5: Projected surface oxygen capacity (saturation concentration) trends: (a) spatial distribution for all CESM ocean grid cells, (b) latitudinal dependence for global coastal points and documented coastal hypoxic areas, and (c) histograms of global coastal points and documented coastal hypoxic areas. The format and color coding follow Fig. 1.**

and documented hypoxic areas are within 8% of the corresponding median oxygen capacity trends. The link between surface oxygen capacity and concentrations diminishes with depth.

The projected global distribution of vertical-minimum oxygen concentration trends indicates decreases in most areas (Fig. 6a). In CESM and in nature, the depth of the vertical-minimum oxygen concentration tends to be near-bottom in coastal areas and

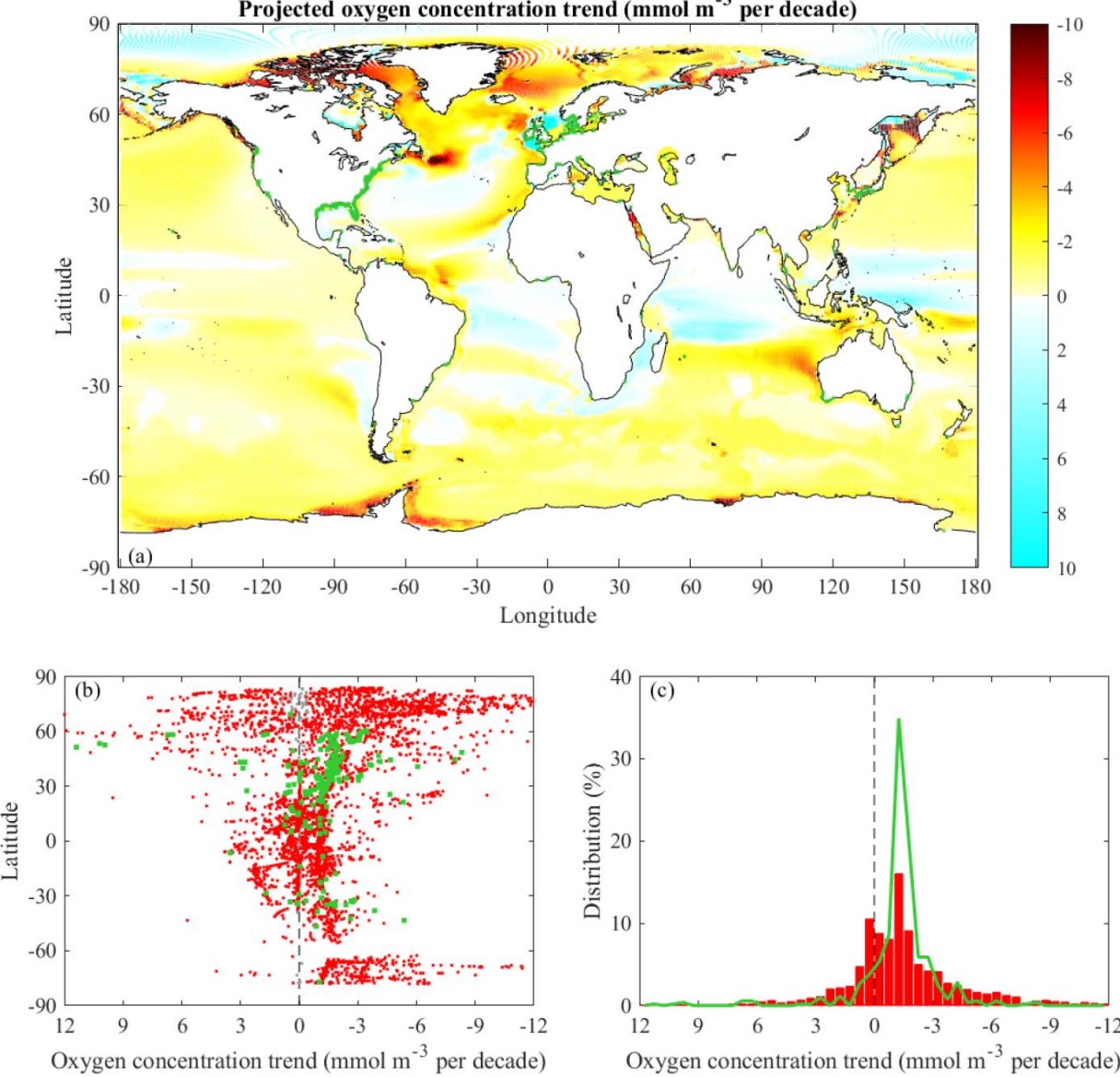

**Figure 6: Projected vertical-minimum oxygen concentration trends: (a) spatial distribution for all CESM ocean grid cells, (b) latitudinal dependence for global coastal points and documented coastal hypoxic areas, and (c) histograms of global coastal points and documented coastal hypoxic areas. The format and color coding follow Fig. 1 except for the blue shading indicating increasing oxygen concentrations.**

is bathymetrically constrained to be relatively close to the surface (within ~100 m), while the minimum oxygen concentration can occur much deeper in the open ocean and consequently have a more remote connection to surface oxygen capacity. The projections for most ocean cells have linear trends with p≤0.10 (Table 1). On average, the $\sigma_s$ is 6% of the local trends. The

global median trend in oxygen is -0.7 mmol m$^{-3}$ per decade and $\sigma_m$ is small (Table 1). The global distribution of vertical-minimum oxygen concentration rates (Fig. 6a) is broadly consistent with the global distribution of the RCP8.5 model-average projected changes in oxygen concentrations (at 200-600 m depth) shown in Bopp et al. (2013). The total ocean oxygen content decrease (from the 1990s to 2090s) calculated in Bopp et al. (2013) translates to a -0.6 mmol m$^{-3}$ per decade trend, which is close to the current results. It is noteworthy that some areas in the tropics and Arctic with projected vertical-minimum oxygen increases in spite of surface oxygen capacity decreases. The interplay between oxygen capacities and concentrations is described below in a coastal context.

The median trend in vertical-minimum oxygen concentrations for global coastal points (-1.2 mmol m$^{-3}$ per decade) is 77% faster than the ocean median rate (Table 1). Similar to oxygen capacity and surface oxygen concentrations, vertical-minimum oxygen concentrations rates tend to be near the median value from 30 ºS to 30 ºN and increase at higher latitudes (exceeding -10.0 mmol m$^{-3}$ per decade), but there is a lot of scatter (Fig. 6b). Similar to the global points, $\sigma_s$ averages only 5% of the trends at coastal points. The median trend for documented hypoxic areas (-1.4 mmol m$^{-3}$ per decade) is somewhat stronger than for all coastal points (Table 1). Histograms (Fig. 6c) indicate most coastal locations (72%) and documented hypoxic areas (89%) have projected vertical-minimum oxygen concentration declines. Some coastal (19%) and documented hypoxic areas (8%) have projected trends faster than -3.0 mmol m$^{-3}$ per decade.

The median trends for vertical-minimum oxygen concentrations for global, coastal, and documented hypoxic points are respectively 49%, 76%, and 98% of the corresponding trends for surface oxygen capacity. The trends in oxygen concentration weaken with depth (for the upper 100-m range analyzed). The difference between rates for vertical-minimum and surface conditions is smallest in documented hypoxic areas where waters generally are shallower than in the coastal and global categories. The differences between trends in vertical-minimum oxygen concentrations and surface oxygen capacity are partially accounted for by the weakening of warming trends with depth (mentioned above) and the temperature-dependence of oxygen capacity. Notwithstanding, the median trends for oxygen concentrations are smaller than the oxygen capacity trends at the corresponding depths. This situation is consistent with decreasing AOU, as indicated by supplementary analysis of AOU trends within the upper 100 m. It should be noted that some coastal areas having stronger, equal, or weaker oxygen rates than oxygen capacity trends and some areas even have oxygen increases despite decreasing oxygen capacity. The ecosystem dynamics for the variety of coastal oxygen situations occurring within the model warrant further investigation beyond this study.

### 3.4 Intra-ensemble variability and context from other projections

Linear trends calculated for each individual member of the CESM RCP8.5 ensemble differ from the ensemble-mean trends already described. The intra-ensemble variability is characterized by $\sigma_e$, the standard error of the ensemble-mean rates of change, which is calculated at each grid cell. Median $\sigma_e$ values for ocean, coastal, and documented hypoxic area are included in Table 2. For SST, and oxygen capacity, and surface oxygen concentration trends, $\sigma_e$ is small locally and the median $\sigma_e$ values are only 5-7% of the median ensemble-mean trends (Table 2). The $\sigma_e$ values are larger for vertical-minimum oxygen

concentration trends. The median $\sigma_e$ is 10-24% of the median ensemble-mean trends; this source of uncertainty is highest in the global ocean and lowest for documented hypoxic areas (Table 2). Overall, the relatively small $\sigma_e$ values indicate that each ensemble member projects long-term trends that are similar to each other and the ensemble-mean projections. Another way to

**Table 2: Variability among CESM RCP8.5 ensemble members for ocean, global coastal points, and documented coastal hypoxic areas where p≤0.10. Median $\sigma_e$ values (over all included grid cells) are reported along with $\sigma_{me}$ values. Corresponding percentages**
**in brackets express the ratio of $\sigma_e$ or $\sigma_{me}$ to the median ensemble-mean trends (Table 1).**

| Rate | Ocean | Ocean | Coastal | Coastal | Hypoxic | Hypoxic |
|---|---|---|---|---|---|---|
| | $\sigma_e$ | $\sigma_{me}$ | $\sigma_e$ | $\sigma_{me}$ | $\sigma_e$ | $\sigma_{me}$ |
| Projected SST (ºC per decade) | 0.02 {6%} | $5 \times 10^{-3}$ {1%} | 0.02 {6%} | $6 \times 10^{-3}$ {2%} | 0.02 {5%} | 0.01 {2%} |
| Projected oxygen capacity (mmol m$^{-3}$ per decade) | 0.08 {7%} | 0.02 {2%} | 0.09 {6%} | 0.05 {3%} | 0.08 {5%} | 0.05 {4%} |
| Projected surf. oxygen conc. (mmol m$^{-3}$ per decade) | 0.08 {6%} | 0.02 {2%} | 0.10 {7%} | 0.03 {2%} | 0.13 {6%} | 0.04 {3%} |
| Projected vert.-min. oxygen conc. (mmol m$^{-3}$ per decade) | 0.15 {24%} | 0.03 {5%} | 0.23 {20%} | 0.03 {3%} | 0.13 {10%} | 0.04 {3%} |

look at the intra-ensemble variability is considering $\sigma_{me}$, the standard deviation of the spatial median rates from each ensemble member (analogous the median values in Table 1). The $\sigma_{me}$ values are only 1-5% of the corresponding median ensemble-mean rates. Thus, the reported median trends are robust regardless of whether an individual run or ensemble-mean is analyzed.

The CESM RCP8.5 ensemble results can be placed within a context of projections from different models with various climate scenarios. The focus is on projections from Earth system models that include biogeochemistry and the ranges of median rates for vertical-minimum oxygen concentrations are compared (Table 3). The contextual focus is on vertical-minimum oxygen concentrations because it is immediately linked with hypoxia and is most demanding of the biogeochemical models. Means and standard deviations of the median rates within each model set are not calculated due to the small number of runs in most
sets. Increasing the number of runs in each set eventually may be practical if results from additional models with biogeochemistry become available on the Earth System Grid Federation. The more moderate CMIP5 RCP4.5 scenario and its updated companion CMIP6 SSP2-4.5 have the smallest rates, particularly in coastal and documented hypoxic areas, but still project appreciable declines in oxygen concentrations. For instance, the CMIP5 RCP4.5 coastal projections are 37-77% the

**Table 3: Projected median linear trends in vertical-minimum oxygen concentrations (mmol m$^{-3}$ per decade) for the ocean, global coastal points, and documented coastal hypoxic areas. All points included in median calculations have linear trends with p≤0.10. Reported numbers are the range of median rates from individual runs included in each model set. The number of models in each set are listed in the count column. The CESM (CMIP5) RCP8.5 set includes the ensemble mean data and all individual ensemble members with oxygen data.**

| Model set | Count | Ocean | Coastal | Hypoxic |
|---|---|---|---|---|
| Other CMIP5 RCP4.5 (HadGEM2-ES, IPSL-CM5A-LR, IPSL-CM5A-MR, MPI-ESM-LR, MPI-ESM-MR) | 5 | -0.77 to -0.26 | -0.88 to -0.43 | -0.74 to -0.34 |
| CESM2 (CMIP6) SSP2-4.5 | 3 | -0.73 to -0.68 | -0.81 to -0.76 | -0.82 to -0.81 |
| Other CMIP5 RCP8.5 (HadGEM2-ES, IPSL-CM5A-LR, IPSL-CM5A-MR, MPI-ESM-LR, MPI-ESM-MR) | 5 | -1.61 to -0.34 | -1.72 to -0.66 | -1.33 to -0.63 |
| **CESM (CMIP5) RCP8.5** | **30** | **-0.80 to -0.65** | **-1.30 to -1.15** | **-1.46 to -1.32** |
| CESM2 (CMIP6) SSP5-8.5 | 3 | -1.02 to -0.94 | -1.52 to -1.50 | -1.65 to -1.63 |
| Other CMIP6 SSP5-8.5 (CanESM5, IPSL-CM6A-LR, MPI-ESM1-2-LR, MPI-ESM1-2-HR) | 4 | -1.74 to -0.71 | -2.25 to -1.18 | -2.00 to -1.13 |


CESM RCP8.5 median coastal trend. The CESM RCP8.5 ensemble members are in the middle range of other CMIP5 RCP8.5 projections for ocean and coastal areas, but are on the high end of rates in documented hypoxic areas. The CESM2 (CMIP6) SSP5-8.5 projected oxygen trends are 13-40% stronger than the CESM RCP8.5 projections. The CESM RCP8.5 and CESM2 SSP5-8.5 projections mostly fall within the low to mid ranges of the CMIP6 SSP5-8.5 projections from other models. The SSP5-8.5 coastal projections from all analyzed models are 103-196% of the CESM RCP8.5 median coastal trend. It is notable that median oxygen trends are not larger in documented hypoxic areas than coastal areas for all projections, though this is the pattern in CESM RCP8.5 and CESM2 SSP5-8.5 runs. Larger median trends in coastal areas than the ocean is a robust pattern for all projections. The projections all point to long-term oxygen declines of order 1 mmol m$^{-3}$ per decade through the 21$^{st}$ century.

**4 Discussion**

The results of this study provide a global perspective on climate warming pressures confronting existing and emerging coastal hypoxic areas. The projected warming and declining oxygen conditions will exert considerable pressure on much of the global coast and current coastal hypoxic areas. Observed and projected rates along coasts are considerably higher than the open ocean. These differences point to the increased climate vulnerability of coastal regions and the need to focus on coastal conditions

separately from open-ocean conditions. Observations indicate the warming and reduced oxygen capacities that coastal waters have been experiencing and the CESM RCP8.5 projection points to even more rapid warming and oxygen declines throughout the 21$^{st}$ century. The new analysis indicates the median warming trend in documented hypoxic areas is much (70-75%) stronger than the observed rate and the median trend reported in Altieri and Gedan (2015) for the moderate A1B emissions scenario. The new results suggest that warming-related pressures likely will accelerate the severity of coastal hypoxia around the world.

The documented hypoxic areas already experience oxygen concentrations at or below the 63 mmol m$^{-3}$ threshold (applied in Diaz and Rosenberg, 2008). If concentrations were at this threshold in 2000, the projected median trend in oxygen capacity

and concentration would represent a 20% reduction by 2100. Furthermore, future oxygen declines of any magnitude will have deleterious ecosystem effects as hypoxia worsens (in intensity and duration) in existing hypoxic areas. Such decreases may bottom out at anoxic conditions in some areas; particularly those locations already experiencing oxygen levels considerably

below the hypoxic threshold. Declines in oxygen conditions along the global coast will create emerging hypoxic areas. Projected warming and oxygen declines are particularly severe at high latitudes in the northern hemisphere. Two adjacent fjords in Norway (Trysfjord and Ofotfjord) are the only sites above the Arctic circle within the documented hypoxic area database (Dommasnes et al., 1994; Diaz et al., 2011), but it is likely new Arctic coastal hypoxic areas will emerge. For instance, recent observations suggest Jago Lagoon, Alaska may be on the brink of hypoxia (Smith, 2012; Beaufort Lagoon Ecosystems

LTER, 2020). High-latitude waters are colder and therefore tend to have higher oxygen capacities farther from the hypoxic threshold, but oxygen conditions are projected to decline most rapidly in these coastal waters. In these high-latitude areas, as in other locations, specific nutrient patterns and local hydrodynamics will play important roles in where hypoxia ultimately develops. Emphasis also should be placed on rapidly growing coastal megacities at low and mid-latitudes, which tend to struggle with wastewater treatment infrastructure as populations increase and are likely to experience emerging or worsening

hypoxia due to such pressures and warming (von Glasow et al., 2013; Varis et al., 2006). It is important to note that ecosystem problems can arise before conditions deteriorate down to the canonical hypoxic threshold (63 mmol m$^{-3}$), as many organisms experience physiological stresses above this threshold and have differing tolerances for low oxygen conditions (Vaquer-Sunyer and Duarte, 2008). In addition to directly reducing oxygen capacities, warming also increases metabolic rates, related biological oxygen demand, and thermal stratification (Brown et al., 2004; Cloern, 2001; Breitburg et al., 2018). Thus, attention

should be paid to all coastal areas with lowering oxygen conditions, not just the areas already experiencing seasonal hypoxia. The projected oxygen declines can erode oxygen gains achieved in systems improved by wastewater treatment and nutrient management. For example, oxygen concentrations in Long Island Sound have risen with reduced nutrient loading after decades of nitrogen management, but hypoxia still occurs and oxygen conditions would have been better if not for warming-related oxygen capacity decreases (Whitney and Vlahos, 2021). Projections made following essentially the same methods as this study

point to warming and deteriorating oxygen conditions that will erode gains made by management (Whitney and Vlahos, 2021). It is noteworthy that the projected rate of oxygen capacity decreases for Long Island Sound is smaller than the trend observed in recent decades. This mismatch may be partially due to the different time periods for the observations and projections and also due to limits in the resolution and dynamics of the CESM RCP8.5 results (Whitney and Vlahos, 2021; discussed further below). In other systems, hypoxia has not decreased in spite of major nutrient management efforts. The Baltic Sea is a well-

studied hypoxic system with large managed reductions in nutrient loads entering from its watersheds. Hypoxic areas, however, have grown in recent decades (Conley et al., 2007; Meier et al., 2019). This study projects warming and decreases in oxygen capacity and concentrations in the Baltic. Projections from a Baltic Sea model point to increasing hypoxia due to warming, increased nutrient loads, and intensified nutrient cycling (Meier et al., 2011). Nutrient loads from the Mississippi watershed feed eutrophication in the Gulf of Mexico coastal hypoxic zone (Scavia et al. 2019; Giudice et al., 2020; Stackpoole et al.,

2021). Despite some managed reductions in the total nitrogen load (Stackpoole et al., 2021; USGS, 2022), hypoxic extent has

not consistently decreased and remains well above the management goal (Rabalais and Turner, 2019). Water temperatures in the area have been rising and projections indicate warming will continue to exert a pressure on oxygen conditions (Turner et al., 2016; Giudice et al., 2020; this study). The projections from this study are consistent with a Gulf of Mexico hypoxia study that points to more severe, prolonged, and extensive hypoxia by the end of the century; primarily due to warming-related

oxygen solubility reductions (Laurent et al., 2018). Hypoxia in the Chesapeake Bay has not been reduced despite extensive nitrogen management and somewhat decreased nitrogen loads (Zhang et al., 2015; Murphy et al., 2011; Zhang et al., 2020; Maryland Department of Natural Resources, 2021; Chesapeake Bay Program, 2022). Moderate oxygen increases tied to load reductions have been overwhelmed by long-term oxygen declines mainly associated with warming (Ni et al., 2020). Projections based on climate downscaling suggest hypoxic volume will increase substantially by mid-century (Ni et al., 2019). Even with

load reduction efforts, nitrogen loads in these areas and many other coastal systems remain much higher than historic levels prior to large increases in human population. In other areas such as the Bohai Sea and Pearl River estuary, nutrient loads have not been reined in by management and hypoxia is worsening in response to increased anthropogenic loads and warming (Qian et al, 2018; Zhai et al., 2019). In general, ameliorating coastal hypoxia through nutrient management has proved challenging. The ongoing and projected warming pressure make efforts to improve coastal oxygen conditions more daunting. Future

management efforts should incorporate projected warming-driven oxygen decreases. It would be wise to consider progressively decreasing maximum nutrient loads to contend with decreasing oxygen capacities.

Projections from CMIP5 and CMIP6 are widely used and provide valuable information for potential climate scenarios. Earth system models, however, have differences in representing oxygen dynamics. This study has a coastal focus, but it is worth noting that some features of global ocean oxygen patterns in models warrant further investigation. For instance, the Arctic is

an area with projected decreases in oxygen capacity and offshore increases in vertical-minimum oxygen concentrations for the CESM RCP8.5 projections. This tendency appears in some of the other CMIP5 models, but not in the CMIP6 SSP5-8.5 projections. The locations and extent of other areas with projected vertical-minimum oxygen increases (mostly in the tropics) vary among models. In most cases, the projected offshore oxygen increases do not reach the coasts. In coastal regions, the vertical-minimum oxygen level is bathymetrically constrained to be closer to the surface and therefore more closely tied to

projected oxygen capacity declines. This factor favors closer agreement in coastal areas among models that have similar warming rates. On the other hand, the relatively coarse resolution and global application of Earth system models make it challenging to represent physical and biogeochemical dynamics in coastal waters. It is encouraging that the new analysis indicates CESM RCP8.5 coastal performance is broadly consistent with observed SST and oxygen capacities during overlapping years. The projected and observed coastal oxygen capacities have a similar latitudinal pattern. These coastal results

indicate more latitudinal pattern agreement between observations and model results than found for open-ocean oxygen concentrations at the thermocline (Oschlies et al., 2017). The nominal 1° latitude CESM resolution, however, offers only a limited representation of coastal processes. Regional scales of variability are resolved, but smaller scales along continental shelves and within estuaries are not. The analysis pairs each documented hypoxic area with the nearest CESM coastal grid point, but most estuaries are not resolved in the CESM grid. Consequently, the projections may not represent oxygen conditions

near the heads of estuaries where hypoxia often occurs. The nearest coastal points reflect conditions in the vicinity of the estuaries. In nature, the influence of surrounding coastal waters on estuarine hypoxic areas is exerted via estuarine exchange (e.g. Kuo et al.,1991; Roegner et al., 2011; Coogan et al., 2021).

Research on future coastal oxygen conditions can be advanced with projections from local high-resolution models (Fennel and Testa, 2019), as for systems such as the Gulf of Mexico (Justic et al., 2007), Chesapeake Bay (Ni et al., 2019), and Baltic Sea (Meier et al., 2019). Such regional or estuary-specific models better resolve processes on continental shelves and within estuaries and often are compared to local datasets for oxygen and biogeochemical variables. Further steps forward will come as Earth system models increase resolution and improve the representation of physical and ecosystem processes in global coastal areas (e.g. Holt et al., 2009; Holt et al., 2017). One way forward is extending the box model approach for estuarine mixing applied in CESM and CESM2 to estuarine biogeochemical cycling (Sun et al., 2017). Other ways forward are nested coastal grids, resolution refinements approaching the coast, or hybrid approaches with higher resolution for some coastal systems and analytical or box model representations in others (Ward et al., 2020). For existing and future Earth system models, the coastal biogeochemical results and representation should be compared to coastal observations and coastal model results where available. Such analyses of biogeochemical performance have been completed for the southwest Pacific and the northwest Atlantic continental shelf; though oxygen is not included in the analyses (Rickard et al., 2016; Laurent et al., 2021). This approach can be extended to other areas and other biogeochemical variables including oxygen.

This study has focused on warming-related pressures on hypoxia, but it is important to note that the development of hypoxia and long-term changes in its prevalence depend on many factors. As described above, anthropogenic increases in nutrient loading and related management efforts play important roles in hypoxic extent and intensity. Interannual and longer-term variability in river flow directly influence terrestrial nutrient loads entering estuaries and other coastal waters. Such changes can be related to climate controls on storm tracks and precipitation (Altieri and Gedan, 2015). In addition, long-term increases in estuary depths due to sea-level rise and stratification due to warming and intensified freshwater inputs can inhibit ventilation of near-bottom waters and increase hypoxia (Cloern, 2001). Overall, the observed and projected deterioration in coastal oxygen conditions are attributable to nutrient overloading fueling eutrophication and anthropogenic climate change (e.g. Rabalais and Turner, 2001; Paerl, 2006; Diaz and Rosenberg, 2011). The present study contributes by describing global coastal distributions of trends in temperature, oxygen capacity, and vertical-minimum oxygen concentration. Future studies should assess the relative importance of the multitude of stressors exacerbating coastal hypoxia, both regionally and globally. Such studies will be facilitated by advances in Earth system models with biogeochemistry.

## 5 Conclusions

A global 40-year observational gridded climate data record (updated from Merchant et al., 2019) is analyzed for linear trends in temperature and oxygen conditions during summer months (August and February averages for northern and southern Hemispheres, respectively). CESM 21[st] century projections for the RCP8.5 scenario are analyzed in similar fashion for summer

months with annual minimum oxygen conditions. The observed and projected trends indicate warming-related pressures on oxygen conditions are increasing. The median projected trends for 2006-2100 along the global coast are 0.39 ºC, -1.6 mmol m$^{-3}$, and -1.2 mmol m$^{-3}$ per decade for SST, surface oxygen capacity (saturation concentration), and vertical-minimum oxygen concentration, respectively. These trends are considerably faster than the median projections for the entire surface ocean. The projected median coastal trends for SST and oxygen capacity are 48% and 18% faster than the corresponding observed rates. Significant portions of the global coast (upwards of 18%) are projected to change even more rapidly, with SST warming more than 0.60 ºC per decade and oxygen capacity and concentrations rates faster than -3.0 mmol m$^{-3}$ per decade. Observed and projected rates tend to increase with latitude. Most (89%) of the documented hypoxic areas (Diaz et al., 2011) are in the mid-latitude (20-60 ºN) northern hemisphere where trends tend to be larger, but not as extreme as coastal rates in the Arctic. The database documents only a few coastal hypoxic areas at high latitudes, where it is likely that new hypoxic areas will emerge due to warming and rapidly deteriorating oxygen conditions. Coastal megacities (at low and mid-latitudes) are likely to experience emerging or worsening hypoxia with the dual pressures of warming and increasing populations. Projections from other models and other climate scenarios (including CIMP5 RCP4.5 and CMIP6 SSP2-4.5 and SSP5-8.5) point to long-term oxygen decreases of order 1 mmol m$^{-3}$ per decade through the 21$^{st}$ century and have larger median trends for coastal waters than for the open ocean. The projected warming and declining oxygen conditions will exert considerable pressure on current hypoxic areas, since future oxygen declines of any magnitude will have deleterious ecosystem effects as hypoxic intensity and duration worsens. The projected oxygen declines can erode oxygen gains achieved in systems improved by nutrient/wastewater management and should be incorporated into coastal environmental management strategies to protect future water quality and ecosystem services.

**Author contribution:** M.W. completed the research and wrote the manuscript.

**Competing interests:** The author declares that he has no conflict of interest.

**Acknowledgements**: This research was supported by the U.S. Environmental Protection Agency Long Island Sound Study (LISS01719).

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
