# Peer review of "Observed and projected global warming pressure on coastal hypoxia"

_Biogeosciences, 2021_

## Author Response (AR1)

**EDITOR**

I have read the comments of both reviewers as well as your replies to their concerns. Both reviewers strongly require further scientific analyses of the presented results to gain understanding. Progresses beyond the current knowledge and interpretations of differences compared to past reference works were also not enough covered. They both missed a sound justification that CESM models with a resolution of 1° (~100 km) can appropriately simulate the coastal zone. A discussion of the role of nutrients is also missing.

Coastal hypoxia mostly occurred on the bottom during the stratification period. The duration and intensity of stratification govern the severity of bottom hypoxia. I am missing a discussion on how the SST correlates with these two characteristics. I also missed a clear definition of the "surface oxygen capacity" and "vertical oxygen minimum". Then, usually the term forecast is reserved for short term prediction a few days to 1-2 weeks (similar to weather forecasts). Here you rather refer to climate projections.

Therefore, I recommend major revisions of your work and would like that you address very clearly all the points (point by point and not a global answer as provided in your reply) raised by the two Reviewers in your answer and revised version. Please provide an annotated revised manuscript clearly highlighting all the changes made. The revised version will be sent for a new round of reviews.

RESPONSE: The paper has been extensively revised following the reviewer comments. New analysis of other CMIP5 and CMIP6 projections for various climate scenarios has been added. This time-intensive effort bolsters the main points. Two new tables are added to quantify intra-ensemble variability (among CESM RCP8.5 ensemble members) and variability among models and climate scenarios. The original table now has uncertainty ranges accompanying the projected trends. The discussion of results and their implications has been greatly expanded. Overall, 34 additional works are cited in the revised manuscript. The paper includes additional description and discussion of the appropriateness of coarse Earth system models for projecting coastal conditions. The role of nutrients now has greater treatment in the Discussion and was already included in the introduction. The paper focus remains on warming-related effects on hypoxia. Additional analysis is now included for temperature and oxygen conditions at different vertical levels. This additional analysis connects surface conditions to bottom conditions in coastal waters.

The "surface oxygen capacity" is the oxygen saturation concentration at the surface of the ocean. The term oxygen capacity is borrowed from physiological literature where it is a standard term (primarily used for oxygen saturation concentration in blood). It has been used previously in oceanographic publications and it is less cumbersome to write "surface oxygen capacity trend" than "surface oxygen saturation concentration trend." Links to saturation concentration are now included in additional places and oxygen capacity now is defined with cited references in the introduction. The term "vertical-minimum oxygen concentration" refers to the minimum oxygen concentration in the water column at each location. This meaning has been clarified in the text. The word "forecast" has been replaced with "projection" or "project" throughout the text as appropriate.

This document follows the suggested format of editor/reviewer comments (highlighted in blue) followed by author responses (highlighted in grey) and corresponding revisions (without highlighting). In some cases the entire section in the paper is referred to rather than including the text here. In most cases, the sentences/paragraphs with revised text are included here to facilitate the subsequent review. The marked-up version of the manuscript indicates all textual changes. Modifications to figures are minor (e.g. changing labels from "forecasted" to "projected" and updating written RMSE value) and are mentioned below. Tables 2 and 3 are new.

REVISIONS: The revisions related to the appropriateness of coarse models in coastal settings and the revisions related to further discussion of the role of nutrients are included in the responses to each reviewer.

Revisions relevant to connections between surface and near-bottom conditions occur in several places:
Section 2.2: Supplementary analysis of temperatures, oxygen concentrations, and AOU at 10-m intervals down to 100 m deep is included to describe transitions from surface conditions to deeper levels in the coastal water column.

Section 3.3: Supplementary trend analysis of temperatures at 10-m intervals down to 100 m deep also indicates robust warming trends (not shown), but the coastal warming rates at the 20-30 m, 40-50 m, and 90-100 m levels decrease to 92%, 80%, and 70% the SST trend magnitude, respectively.

Section 3.3: The distribution of surface oxygen concentration trends (not shown) is very similar in terms of magnitudes and spatial patterns to the distribution for oxygen capacity trends (Figure 5). The median surface oxygen concentration trends for global, coastal, and documented hypoxic areas are within 8% of the corresponding median oxygen capacity trends. The link between surface oxygen capacity and concentrations diminishes with depth.

Section 3.3: The trends in oxygen concentration weaken with depth (for the upper 100-m range analyzed). The weakening of warming trends with depth (mentioned above) accounts for part of this difference due to the temperature-dependence of oxygen capacity. Notwithstanding, the median trends for oxygen concentrations are smaller than the oxygen capacity trends at the corresponding depths. This situation is consistent with decreasing AOU, as indicated by supplementary analysis of AOU trends within the upper 100 m.

The term "oxygen capacity" now is more clearly defined and linked to oxygen saturation concentration throughout the text:
Abstract: oxygen saturation concentration at the surface (surface oxygen capacity)

Introduction: Oxygen saturation concentration (oxygen capacity) decreases with increased water temperatures (Weiss, 1970; Garcia and Gordon, 1992), which can exacerbate hypoxia. Oxygen capacity is a succinct synonym for oxygen saturation concentration which is borrowed from physiological research on blood oxygen levels (e.g. Haldane and Smith, 1900; Black, 1940; Maio and Neville, 1965; Bernal et al., 2018) and has been applied to dissolved oxygen in the coastal and open ocean (Helm et al., 2011; Deignan-Schmidt and Whitney, 2017). Metabolic rates and related oxygen demands also rise with temperature (Brown et al., 2004).

Section 2.1: Oxygen capacities are the oxygen saturation concentrations calculated with the Garcia and Gordon (1992) equations

Section 2.2: surface oxygen capacity (saturation concentration at the surface, the "O2SAT" variable)

Conclusions: surface oxygen capacity (saturation concentration)

The term "vertical-minimum oxygen concentration" is now described in two places within the text:

Introduction: The main objective of this paper is to quantify global patterns exacerbating coastal hypoxia by analyzing linear trends in SST, surface oxygen capacity, and vertical-minimum oxygen concentrations (the minimum dissolved oxygen in the water column at each location).

Section 2.2: vertical-minimum oxygen concentration (minimum dissolved oxygen concentration in each water column, the "O2_ZMIN" variable)
All occurrences of "forecast" or related forms have been changed to "projection" or "projected."

**REVIEWER 1**

Summary
This manuscript analyzed the long-term trend of SST and surface oxygen capacity from 40-year global gridded climate data record from a satellite platform compared it with CMIP5 CESM Large Ensemble mean on the simulated trend. The forecasted median trend under RCP 8.5 forcing along the global coast are 0.39deg, -1.6 mmol m$^{-3}$, and 1.2 mmol m$^{-3}$ per decade for SST, surface oxygen capacity and vertical-minimum oxygen concentration, respectively. The trends in the forecasted global coastal region are much faster than the median rate for the entire ocean, and they are also much faster than the corresponding observed rates. This study also highlighted that warming and oxygen decline rate are larger at high latitude that it may cause new emerging hypoxic areas.

The manuscript will benefit the field of research in global deoxygenation due to warming by providing an estimate of expected changes in SST, sea surface oxygen solubility and vertical oxygen minimum in the global context in the future. However, this paper is a little thin on content and present numbers like a report. It lacks of more advanced understanding, deeper analysis on this topic, and comparison with previous similar study. Although I see the value of this work, I perceive that the publication is premature at this time. My major and detailed comments are listed as below.

RESPONSE: I appreciate the reviewer's comments. The concerns and questions raised have been addressed with substantial additional analysis and a greatly expanded discussion (as described below). There is a clear need for updated 21$^{st}$ century forecasts for conditions in coastal hypoxic areas around the world. The paper presents and describes such forecasts and places them in context with global observations. The revised paper more effectively motivates the objectives and much more fully discuss results, limitations, implications, and ways to improve forecasts (as described below). This paper should inform many scientists and managers about the intensity and spatial distribution of warming pressures confronting coastal hypoxic areas.

Major comments:
(1) This manuscript missed a more in-depth discussion and insight of causes on the observation/model simulation comparison and global/coastal/hypoxic region comparison. Some interpretations of data are debatable. For example, the median trend of vertical oxygen minimum concentration decline in hypoxic areas is faster than all the global points (Table1, L275 and L282). Why is it due to little coverage of high latitude? Based on Figure 6(a), there are regions nearly the north pole with a large increase in oxygen concentration, which also missed further discussion in the manuscript. From the review's perspective, there are potential questions that could be further explored and discussed. For example, why the forecasted hypoxic region has a faster vertical minimum oxygen decline rate than the coastal region, while the forecasted oxygen capacity decreases at slower rate in the hypoxic region? Why the forecasted SST increases faster in the hypoxic region than the coastal zone, which is the opposite from the observation???

RESPONSE: More in-depth analysis and discussion of results in terms of the observation/model simulation comparison and the global/coastal/hypoxic region has been added. The helpful questions posed by the reviewer helped guide the expanded analysis and discussion. The revised paper delves more deeply into such points and includes extensive new analysis as described below.  More detailed responses are included below in connection to other detailed reviewer comments.

REVISIONS: See revisions detailed below in connection to specific reviewer comments. The extensive revisions occur throughout the paper.

(2) Second, this manuscript lacks further analysis regarding the discrepancy/disagreement with the previous study and discussion about the possible causes. It also didn't include any uncertainty analysis, like what are the pro and cons of using CESM large ensemble rather than using multiple different GCMs? What is new about this study compared to previous similar papers (i.e. Gilbert et al.,2010; Bopp et al., 2013)? For the disagreement in estimated rate, the observed global median oxygen capacity trend is -0.9 mmol m$^{-3}$ per decade, several times faster than Gilbert et al. (2010) for 1976-2000. What are the primary causes for the difference? The different time periods covered? The different data sources? Or other reasons? Similarly, for the forecasted trend, there are also discrepancies from Bopp et al. (2013). From the author's perspective, which one is more credible?

RESPONSE:
Uncertainty analysis for regressions has been added to the text and tables. Comparisons among the CESM RCP8.5 ensemble members have been added (including a new table).

Extensive new analysis comparing the primary results to projections from other CMIP5 and CMIP6 models and scenarios has been added for context. This was a major undertaking that bolstered the papers main points. New sections have been added to the paper. The revisions are described immediately below and later connected to another specific reviewer comment.

There are many reasons for differences between this study and prior studies including the different time periods, areas, and depths analyzed and other differences in study designs. These differences and the underlying reasons will be discussed in more detail in light of the questions the reviewer poses. Consequently, a perfect match is not expected, but the prior studies provide context. No value judgement is made on which studies are more credible. These differences and the underlying reasons are included in the revised text (as described below in response to the reviewer's specific comments on these studies).

REVISIONS:
Section 2.4: Uncertainty associated with the regression fit is quantified with $\sigma_s$, the standard error of the regression slope. Based on the $\sigma_s$ values, the corresponding standard error of the spatial median values ($\sigma_m$) are reported for global, coastal, and documented hypoxic areas. For CESM RCP8.5 projections, variability among ensemble members is characterized by calculating regression slopes for each member and then calculating the associated standard error of the ensemble-mean rates of change ($\sigma_e$) at each grid cell. The spatial median rates for each ensemble member also are calculated and the corresponding standard deviation of these median rates ($\sigma_{me}$) are reported as another way to characterize intra-ensemble variability. Rates of change are shown for the entire global ocean and emphasis is placed on change in coastal waters.

Section 2.6: A new section describing the methods for comparisons with projections from different models and climate scenarios.

Table 1: Median rates are listed with ±$\sigma_m$, the standard errors of the median values.

Section 3.1: The $\sigma_s$ and $\sigma_m$ for each variable are described in the text.

Section 3.3: The $\sigma_s$ and $\sigma_m$ for each variable are described in the text.

Section 3.4: See the new section on "Intra-ensemble variability and context from other projections" that includes uncertainty analysis.

Table 2: New table on variability among ensemble members.

Table 3: New table with projections from different models and climate scenarios.

Discussion: This study has a coastal focus, but it is worth noting that some features of global ocean oxygen patterns in models warrant further investigation. For instance, the Arctic is an area with projected decreases in oxygen capacity and offshore increases in vertical-minimum oxygen concentrations for the CESM RCP8.5 projections. This tendency appears in some of the other CMIP5 models, but not in the CMIP6 SSP5-8.5 projections. The locations and extent of other areas with projected vertical-minimum oxygen increases (mostly in the tropics) vary among models. In most cases, the projected offshore oxygen increases do not reach the coasts. In coastal regions, the vertical-minimum oxygen level is bathymetrically constrained to be closer to the surface and therefore more closely tied to projected oxygen capacity declines. This factor favors closer agreement in coastal areas among models that have similar warming rates.

Conclusions: Projections from other models and other climate scenarios (including CIMP5 RCP4.5 and CMIP6 SSP2-4.5 and SSP5-8.5) point to long-term oxygen decreases of order 1

Added text describing differences in methodologies and results for prior studies such as Gilbert et al. (2010) and Bopp et al. (2013) are detailed below in response to the reviewer's specific comments on these studies.

(3) Except for the hypoxia in larger coastal systems like the Northern Gulf of Mexico, Baltic Sea, East China Sea, etc., the majority of coastal hypoxia occurs in small scale estuarine systems, which are not able to be covered, or could not be represented by the Global Earth System models. The CESM model with resolution at 1deg could not accurately represent both the physical and biogeochemical dynamics. Therefore, I am much concerned about the results on the forecasted vertical oxygen minimum in the hypoxic region. For example, I am suspicious about the result in Table 1 on the forecasted change rate of surface oxygen capacity and vertical oxygen minimum in the hypoxic region. In the coastal hypoxic system like the Chesapeake Bay and the Gulf of Mexico, the bottom minimum oxygen concertation region was generally much smaller than 62.5 mmol m$^{-3}$ and the anoxia existed. In this case, there should be barely room for further oxygen decline due to warming. Thus, the decrease of surface oxygen capacity could be greater.

RESPONSE: There are concerns (shared by the author) about the applicability of forecasts from global climate models to estuarine systems. Nevertheless, this approach has been used in previous studies which have been cited by many. The limitations are described in the introduction. The revised paper discusses these limitations in more detail. The discussion also highlights the need for long-term forecasts with high resolution regional or estuary-specific models. There are some cited examples where such modeling has been completed, but they represent a small portion of the world's estuaries and coastal systems.

[revised manuscript text omitted]

(4) One big component missing in the discussion of the entire manuscript is nutrients. Unlike the open ocean oxygen minimum zone, the nutrient load into the coastal waters is the determining factor for hypoxia development. Although the rising temperature will reduce the oxygen solubility in the water and cause oxygen concentration decline, the hypoxia formation is also subjected to the local hydrodynamics (e.g. stratification), riverine nutrient loading and larger-scale ocean circulation. The ecological system (e.g. phytoplankton-zooplankton-bacteria coupling) may also shift with changing temperature. Thus, the rising temperature might not necessarily lead to the expansion of hypoxic water. The above points are worthy to discuss in the manuscript.

RESPONSE: The reviewer suggests that the important role of nutrients on hypoxia should be added to the discussion. The paper focuses on the warming-related pressures and alludes to the nutrient role early on. A discussion on nutrients and nutrient management has been added.

REVISIONS:
Discussion: The projected oxygen declines can erode oxygen gains achieved in systems improved by wastewater treatment and nutrient management. For example, oxygen concentrations in Long Island Sound have risen with reduced nutrient loading after decades of nitrogen management, but hypoxia still occurs and oxygen conditions would have been better if not for warming-related oxygen capacity decreases (Whitney and Vlahos, 2021). Projections made following essentially the same methods as this study point to warming and deteriorating oxygen conditions that will erode gains made by management (Whitney and Vlahos, 2021). It is noteworthy that the projected rate of oxygen capacity decreases for Long Island Sound is smaller than the trend observed in recent decades. This mismatch may be partially due to the different time periods for the observations and projections and also due to limits in the resolution and dynamics of the CESM RCP8.5 results (Whitney and Vlahos, 2021; discussed further below). In other systems, hypoxia has not decreased in spite of major nutrient management efforts. The Baltic Sea is a well-studied hypoxic system with large managed reductions in nutrient loads entering from its watersheds. Hypoxic areas, however, have grown in recent decades (Conley et al., 2007; Meier et al., 2019). This study projects warming and decreases in oxygen capacity and concentrations in the Baltic. Projections from a Baltic Sea model point to increasing hypoxia due to warming, increased nutrient loads, and intensified nutrient cycling (Meier et al., 2011). Nitrogen loads have been reduced in the Mississippi watershed, which feeds eutrophication in the Gulf of Mexico coastal hypoxic zone (Scavia et al. 2019; Giudice et al., 2020; USGS, 2022). Despite the nitrogen management, hypoxic extent has not consistently decreased and remains well above the management goal (Rabalais and Turner, 2019). Water temperatures in the area have been rising and projections indicate warming will continue to exert a pressure on oxygen conditions (Turner et al., 2016; Guidice et al., 2020; this study). The projections from this study are consistent with a Gulf of Mexico hypoxia study that points to more severe, prolonged, and extensive hypoxia by the end of the century; primarily due to warming-related oxygen solubility reductions (Laurent et al., 2018). Hypoxia in the Chesapeake Bay has not been reduced despite extensive nitrogen management and somewhat decreased nitrogen loads (Murphy et al., 2011; Maryland Department of Natural Resources, 2021; Chesapeake Bay Program, 2022). Moderate oxygen increases tied to load reductions have been overwhelmed by long-term oxygen declines mainly associated with

warming (Ni et al., 2020). Projections based on climate downscaling suggest hypoxic volume will increase substantially by mid-century (Ni et al., 2019). In other areas such as the Bohai Sea and Pearl River estuary, nutrient loads have not been reined in by management and hypoxia is worsening in response to increased anthropogenic loads and warming (Qian et al, 2018; Zhai et al., 2019). In general, ameliorating coastal hypoxia through nutrient management has proved challenging. The ongoing and projected warming pressure make efforts to improve coastal oxygen conditions more daunting. Future management efforts should incorporate projected warming-driven oxygen decreases. It would be wise to consider progressively decreasing maximum loads to contend with decreasing oxygen capacities.

Discussion: This study has focused on warming-related pressures on hypoxia, but it is important to note that the development of hypoxia and long-term changes in its prevalence depend on many factors. As described above, anthropogenic increases in nutrient loading and related management efforts play important roles in hypoxic extent and intensity. Interannual and longer-term variability in river flow directly influence terrestrial nutrient loads entering estuaries and other coastal waters. Such changes can be related to climate controls on storm tracks and precipitation (Altieri and Gedan, 2015). In addition, long-term increases in estuary depths due to sea-level rise and stratification due to warming and intensified freshwater inputs can inhibit ventilation of near-bottom waters and increase hypoxia (Cloern, 2001). Overall, the observed and projected deterioration in coastal oxygen conditions are attributable to nutrient overloading fueling eutrophication and anthropogenic climate change (e.g. Rabalais and Turner, 2001; Paerl, 2006; Diaz and Rosenberg, 2011). The present study contributes by describing global coastal distributions of trends in temperature, oxygen capacity, and vertical-minimum oxygen concentration. Future studies should assess the relative importance of the multitude of stressors exacerbating coastal hypoxia, both regionally and globally. Such studies will be facilitated by advances in Earth system models with biogeochemistry.

(5) The oxygen loss from ocean with rising temperature and getting more severe towards higher latitude is pretty much predictable according to the nature of oxygen gas. From this point, this paper did not provide any innovative insight on the topic of global warming pressure on coastal hypoxia. More intriguing questions should be like: how was the impact of the warming pressure compared to the nutrient management strategies to reduce hypoxia? Will it completely overturn the mitigation of coastal hypoxia from nutrient load reduction? Another suggestion is to add regional case studies in some large coastal hypoxic system (e.g. Baltic Sea, Northern Gulf of Mexico, Gulf of St. Lawrence on the high latitude, etc.) combined with the analysis of this study might help to prove.

RESPONSE: The reviewer points out that the oxygen loss from the ocean with rising temperature and getting more severe towards higher latitude is predictable according to the nature of oxygen gas. This is a fair point, but it is important to quantify the forecasted changes and patterns as is presented in this paper. These results are very useful to many people studying and contending with hypoxia issues in a warming climate. The reviewer suggested potentially answering questions about how warming pressure compares to gains made by nutrient management strategies and/or treating case studies of particular coastal hypoxic systems. Refocusing the paper in these ways is beyond the scope of the revisions that will be submitted. However, these issues are now treated in the Discussion. Text has been added on nutrient management, oxygen trends, and warming pressures for the Long Island Sound, Baltic Sea, Gulf of Mexico, Chesapeake, Bohai Sea, and Pearl River estuary. The overall point is made that warming-related pressures make the challenge of improving

coastal oxygen levels more daunting, particularly since current nutrient management efforts have already to struggled to achieve targeted gains.

REVISIONS: Revisions are included above in response to the previous comment.

Detailed comments:

L36: Here the authors noted that this paper aimed to update the analysis with more recent climate modeling. However, it didn't provide any discussion on the model reliability/uncertainty analysis among different GCMs and RCP/A1B emission scenarios, or an explanation on the discrepancy with the previous study.

RESPONSE: Extensive new analysis of other models is now included for context. This represents a large time investment that helped emphasize the main points. These models include different models for the RCP8.5 scenario and other climate scenarios for CMIP5 (RCP4.5) and CMIP6 (SSP2-4.5 and SSP5-8.5). Text, new sections, and a table have been added to the manuscript to address this point:

REVISIONS:

Abstract: Companion analysis of other models and climate scenarios indicates projected coastal oxygen trends for the more moderate RCP4.5 and updated SSP5-8.5 scenarios respectively are 37-77% and 103-196% of the CESM RCP8.5 projections.

Introduction: Companion analysis of other CMIP5 RCP8.5 models, the more moderate RCP4.5 scenario, and the corresponding updated CMIP6 Shared Socioeconomic Pathways (SSP) 2-4.5 and 5-8.5 provides context for the CESM RCP8.5 coastal results.

Methods: See new Section 2.6 "Other Projections"

Results: See new Section 3.4 "Intra-ensemble variability and context from other projections"

Table 3: New table with projections from different models and climate scenarios.

Discussion: Projections from CMIP5 and CMIP6 are widely used and provide valuable information for potential climate scenarios. Earth system models, however, have differences in representing oxygen dynamics. This study has a coastal focus, but it is worth noting that some features of global ocean oxygen patterns in models warrant further investigation. For instance, the Arctic is an area with projected decreases in oxygen capacity and offshore increases in vertical-minimum oxygen concentrations for the CESM RCP8.5 projections. This tendency appears in some of the other CMIP5 models, but not in the CMIP6 SSP5-8.5 projections. The locations and extent of other areas with projected vertical-minimum oxygen increases (mostly in the tropics) vary among models. In most cases, the projected offshore oxygen increases do not reach the coasts. In coastal regions, the vertical-minimum oxygen level is bathymetrically constrained to be closer to the surface and therefore more closely tied to projected oxygen capacity declines. This factor favors closer agreement in coastal areas among models that have similar warming rates.

Conclusions: Projections from other models and other climate scenarios (including CIMP5 RCP4.5 and CMIP6 SSP2-4.5 and SSP5-8.5) point to long-term oxygen decreases of order 1 mmol m$^{-3}$ per decade through the 21st century and have larger median trends for coastal waters than for the open ocean.

L57: The linear trend analysis was applied throughout the manuscript, but the trend/longterm change is not necessarily linear (i.e. quadratic or exponential, etc.)

RESPONSE: In general this is true, but the results show the linear trends work well in most areas (as indicated by the regression statistics) and the focus is on projecting the long-term rates characterizing the rest of the 21st century. A sentence describing this has been added.

REVISIONS: Section 2.4: Linear regression analysis is applied to characterize long-term temporal trends. It is worth noting that nonlinear trends are possible, but the strategy is to focus on constant long-term rates of change and the results indicate linear trends fit the data well in most locations globally.

L58: how does the satellite-derived SST compare to in-situ measurement? Is there any bias?

RESPONSE: The SST product is designed to represent near-surface in-situ measurements. This point is now included in the Methods.

REVISIONS: Section 2.1: The global observational dataset analyzed is the satellite-based SST time-series described in Merchant et al. (2019) and available with updates at the Climate Data Store of the Copernicus Climate Change Service (Embury and Good, 2021). The Level-4 (version 2.0) product combines SST data from several satellite platforms to construct a high-quality climate data record that has been validated with *in situ* observations.

L60-61: using one GCM (CESM) may lead to bias in the projection

RESPONSE: As described above, extensive new analysis of other models and climate scenarios has been added.

REVISIONS: See revisions described above.

L78: using constant 35 salinity will lose the impact of changes in salinity due to circulation and freshwater flow discharge. Any justifications?

RESPONSE: It is true that using a constant salinity removes any impacts from salinity changes. Such impacts, however, are smaller than the warming-related changes for the range of likely salinity trends. Text covering this point has been added in the Methods and sensitivity calculations have been added to the Results. The choice of a constant salinity does not lead to much error in the oxygen capacity trends for a range of long-term salinity changes.

REVISIONS:

Section 2.1: Oxygen capacities are the oxygen saturation concentrations calculated with the Garcia and Gordon (1992) equations using the monthly averaged SST data and a constant 35 salinity. The constant salinity is used because the Merchant et al. (2019) product does not include salinity and because this straightforward approach is sufficient to provide observational context for the projections. The sensitivity of oxygen capacity trends to the choice of salinity is assessed with different salinities specified in the calculations.

Section 3.1: As described in the methods, oxygen capacity is calculated from SST and a constant 35 salinity. This approach neglects changes in oxygen capacity due to long-term salinity variability and the constant salinity choice may bias the calculate oxygen capacity

trends. Sensitivity calculations with constant salinity values of 32 and 34 had RMSE (relative to the original calculations) of at most $10^{-3}$ mmol m$^{-3}$ per decade. Introducing a long-term linear salinity trend of 34.9 to 35.1 (or vice versa) over the observation period created similarly small RMSE of $3 \times 10^{-3}$ mmol m$^{-3}$ per decade. The sensitivity calculations indicate that assuming a constant salinity does not introduce much error in the oxygen capacity trends.

L104-107: large ensemble mean of climate projection is not suitable for short-term point-to-point comparison since it loses decadal variability in the observation

RESPONSE: The temporal mean values for the 2006-2021 period are calculated from the observations and model results. It is reasonable to compare the mean values to look for model bias relative to the observations. As noted in the comment and in the paper, comparing yearly time series over this relatively short 16-year period is not particularly useful. This reasoning is already included at the end of the paragraph. Text has been added emphasizing that mean values are compared.

REVISIONS:

Section 2.3: The summer month values (calculated as described in previous section) for the overlapping period are averaged together to determine mean observed and projected values at each coastal point. The resulting mean SST and oxygen capacity values are used to assess local and global biases relative to observations.

Section 3.2: The observational SST record is compared to CESM RCP8.5 projected coastal conditions for the overlapping 16 years spanning 2006-2021 (Fig. 3a). As described in the methods, the comparison involves mean summer-month values for the overlapping period rather than comparing time series with relatively short-term interannual variability.

L111-112: compare the seasonal average should be better since the annual mean might conceal the Tmax/Omin information

RESPONSE: Annual means are not compared; summer values are. It is a good point that annual means would conceal the Tma/Omin information; that is why summer values are compared.

REVISIONS: No changes needed.

L131: In an opposite opinion, those areas should be included and highly possible to become hypoxic with future warming temperature since nutrient is a big part of coastal hypoxia

RESPONSE: This is a fair point. The eutrophic locations in the database are included within the total set of global coastal points that are analyzed. The text has been modified to express this.

REVISIONS: The Diaz et al. (2011) database also includes 244 additional locations classified as eutrophic (but not hypoxic); these documented eutrophic points are not isolated as a group in this analysis, but are included among the global coastal points analyzed.

L170, 173-174: why the global rate is much faster than other literature?

RESPONSE: The time period and water depths are not the same for the prior study; so an exact match is not expected. The prior study, nevertheless, provides context. Text has been added conveying these points.

REVSIONS:

Section 3.1: The calculated global median rate is several times faster than the median rate of -0.2 mmol m$^{-3}$ per decade observed in offshore (>100 km from coast) upper-ocean (0-300 m) waters for 1976-2000 (Gilbert et al., 2010). The mismatch with the prior study is likely due to the more recent time period and the reliance on surface, rather than upper-ocean, observations. Due to these methodological differences, matching rates between studies is not expected, but the earlier study does provide context.

Section 3.1: The observed median coastal rate is half of the rate calculated for a global coastal band (within 30 km of the coast) for 1976-2000 (Gilbert et al., 2010). For the reasons mentioned above, a match between the studies is not expected. It is interesting that the current study has faster global rates and slower coastal rates; the underlying reasons are not explored here.

L165, 172: there is an issue in rounding from Table 1

RESPONSE: Thank you for catching this. The numbers in the text have been corrected.

REVISIONS:

Section 3.1: The observed global median oxygen capacity trend at the surface (including only points with p≤0.10) is -0.8 mmol m$^{-3}$ per decade and σm is very small (Table 1).

Section 3.1: The observed median oxygen capacity trend for global coastal points (-1.3 mmol m$^{-3}$ per decade, for points with p≤0.10) is 62% faster than the surface ocean median rate (Table 1).

L184-185: why the differences in scatters reach up to 5deg, while RMSE is just 0.03?

RESPONSE: Thank you for catching this. The wrong statistic was reported for RMSE, which is much larger and consistent with the scatter. The text and figure have been corrected.

REVISIONS:

Figure 3: Corrected RMSE values shown on figure.

Section 3.2: The projected temperatures have a small positive bias (0.3 °C) and a moderate RMSE (1.9 °C) relative to observations.

Section 3.2: Projected oxygen capacities have a bias and RMSE of -5 and 17 mmol m$^{-3}$, respectively.

L210-212: why this study generates greater warming forecast than previous study?

RESPONSE: The difference is due to using multiple models and different time periods. This is now described in the text.

REVISIONS: Section 3.3: The projected global median SST trend is 0.35 °C per decade and the associated σm is negligible (Table 1). Global distributions of SST warming have been studied in detail for multiple models and RCP scenarios (e.g. Bopp et al., 2013). Bopp et al. (2013) includes CESM simulations in an analysis of ten models running the RCP8.5 scenario and finds the global average SST increase is 0.27 °C per decade (from the 1990s to 2090s) when averaged across all included models. The smaller warming rate for the Bopp et al. (2013) results is connected to including multiple models and due to calculating rates relative

to the 1990-1999 historical period instead of limiting analysis to the 2006-2100 CMIP5 projection period.

L224-225: P>0.1??? I don't understand this sentence

RESPONSE: The sentence had an error: "low" was written where it should have said "high." This error has been fixed. The meaning is the p-values are high for observations throughout the Southern Ocean so there is no reliable observed trend to compare projections to.

REVISIONS: The high p-values of observed SST trends in much of the Southern Ocean (p>0.10) preclude comparisons of projected and observed spatial structure in this region.

Figure 4c, 5c: why the distribution of hypoxic SST/capacity trend is similar to coastal SST/capacity in the forecast, which is different from observation?

RESPONSE: This question is unclear to me, but it is true that the histograms in Figures 3c, 4c, and 5c are different. The differences between the observations and projections are described in the text.

REVISIONS: None made.

Table 1: forecasted oxygen capacity rate is larger than forecasted oxygen concentration rate in the global and coastal ocean, what are the reasons?

RESPONSE: Supplementary analysis of temperatures and oxygen conditions at different depths has been added to compare and contrast surface and sub-surface conditions. The differences mentioned are described now when discussing the vertical-minimum oxygen concentration trends. Part of the reason is the temperature trends become smaller at depth and correspondingly the oxygen capacity trends are weaker at depth. The other reason is the oxygen concentrations decrease less rapidly than oxygen capacity at the same depth. The underlying reasons should be explored in another study investigating the ecosystem dynamics within the model with a particular emphasis on coastal areas. Text describing the supplementary analysis has been added to the Methods and Results. Note that the revisions are included above in the response to the Editor's comments.

Section 2.2: Supplementary analysis of temperatures, oxygen concentrations, and AOU at 10-m intervals down to 100 m deep is included to describe transitions from surface conditions to deeper levels in the coastal water column.
Section 3.3: Supplementary trend analysis of temperatures at 10-m intervals down to 100 m deep also indicates robust warming trends (not shown), but the coastal warming rates at the 20-30 m, 40-50 m, and 90-100 m levels decrease to 92%, 80%, and 70% the SST trend magnitude, respectively.
Section 3.3: The distribution of surface oxygen concentration trends (not shown) is very similar in terms of magnitudes and spatial patterns to the distribution for oxygen capacity trends (Figure 5). The median surface oxygen concentration trends for global, coastal, and documented hypoxic areas are within 8% of the corresponding median oxygen capacity trends. The link between surface oxygen capacity and concentrations diminishes with depth.
Section 3.3: The trends in oxygen concentration weaken with depth (for the upper 100-m range analyzed). The weakening of warming trends with depth (mentioned above) accounts for part of this difference due to the temperature-dependence of oxygen capacity. Notwithstanding, the median trends for oxygen concentrations are smaller than the oxygen capacity trends at the corresponding depths. This situation is consistent with decreasing AOU, as indicated by supplementary analysis of AOU trends within the upper 100 m.

Figure 6b: what lead to the oxygen increase in some coastal regions when the oxygen capacity decrease?

RESPONSE: As mentioned above, supplementary analysis now is included to describe the differences between surface oxygen capacity trends and vertical-minimum oxygen concentrations trends. There are some areas with increasing oxygen concentrations, though most have decreasing oxygen. The underlying reasons require a detailed dynamic analysis of ecosystem dynamics within the model. Such analysis would require a future study. This is now mentioned.

REVISIONS: See revisions described above and additionally in Section 3.3: It should be noted that some coastal areas having stronger, equal, or weaker oxygen rates than oxygen capacity trends and some areas even have oxygen increases despite decreasing oxygen capacity. The ecosystem dynamics for the variety of coastal oxygen situations occurring within the model warrant further investigation beyond this study.

**REVIEWER 2**

The global analysis of SST and oxygen declines in coastal areas on a global scale is relevant to the persistent issue of oxygen depletion and helps contrast warming effects on the coastal ocean from the open ocean. Although the paper is relatively clearly written, and makes a couple of relevant points about more rapid coastal warming and vulnerability in the far northern hemisphere, I found the paper to lack a detailed discussion of the results that would make the findings compelling. It is really a descriptive summary of the SST and oxygen changes, and almost no mechanistic insights are gained. I include some "minor edits" at the end of this review, but immediately below I try and articulate the much bigger issues and where I think the paper must expand or provide more detail and analysis to be a new contribution to the literature.

RESPONSE: There is a clear need for updated 21$^{st}$ century forecasts for conditions in coastal hypoxic areas around the world. The paper presents and describes such forecasts and places them in context with global observations. The concerns and questions raised are addressed with additional analysis and expanded discussion of results in light of both reviewer's comments. The revised paper effectively motivates the objectives and much more fully discuss results, limitations, implications, and ways to improve forecasts. The extensive new analysis of other climate model results took considerable time and helped bolster the main points of the paper. This paper should inform many scientists and managers about the intensity and spatial distribution of warming pressures confronting coastal hypoxic areas. Responses to specific comments and corresponding revisions are detailed below.

(1) Because only the RCP8.5 high emissions scenario was used, I found the analysis to lacking in its representation of multiple future possible outcomes for oxygen. Might the observed trend be similar to the forecasted trend if another, less high emissions scenario was used? Can the sensitivity of the results to a different scenario be included to expand the scope of the analysis?

RESPONSE: Results for other Earth System Models and other climate scenarios (from CMIP5 and CMIP6) were accessed via the Earth System Grid Federation and analyzed to provide context for the CESM RCP8.5 projections. Data retrieval, processing, and analysis was a time-intensive major undertaking. New sections has been added to the Methods and Results. Text has been added to other sections and a new table has been added. The comparisons add a new facet to the paper.

REVISIONS:

Abstract: Companion analysis of other models and climate scenarios indicates projected coastal oxygen trends for the more moderate RCP4.5 and updated SSP5-8.5 scenarios respectively are 37-77% and 103-196% of the CESM RCP8.5 projections.

Introduction: Companion analysis of other CMIP5 RCP8.5 models, the more moderate RCP4.5 scenario, and the corresponding updated CMIP6 Shared Socioeconomic Pathways (SSP) 2-4.5 and 5-8.5 provides context for the CESM RCP8.5 coastal results.

Methods: See new Section 2.6 "Other Projections"

Results: See new Section 3.4 "Intra-ensemble variability and context from other projections"

Table 3: New table with projections from different models and climate scenarios.

Discussion: Projections from CMIP5 and CMIP6 are widely used and provide valuable information for potential climate scenarios. Earth system models, however, have differences in representing oxygen dynamics. This study has a coastal focus, but it is worth noting that some features of global ocean oxygen patterns in models warrant further investigation. For instance, the Arctic is an area with projected decreases in oxygen capacity and offshore increases in vertical-minimum oxygen concentrations for the CESM RCP8.5 projections. This tendency appears in some of the other CMIP5 models, but not in the CMIP6 SSP5-8.5 projections. The locations and extent of other areas with projected vertical-minimum oxygen increases (mostly in the tropics) vary among models. In most cases, the projected offshore oxygen increases do not reach the coasts. In coastal regions, the vertical-minimum oxygen level is bathymetrically constrained to be closer to the surface and therefore more closely tied to projected oxygen capacity declines. This factor favors closer agreement in coastal areas among models that have similar warming rates.

Conclusions: Projections from other models and other climate scenarios (including CIMP5 RCP4.5 and CMIP6 SSP2-4.5 and SSP5-8.5) point to long-term oxygen decreases of order 1 mmol m$^{-3}$ per decade through the 21st century and have larger median trends for coastal waters than for the open ocean.

(2) The model forecasts that are used are based on a relatively coarse global model. I can't tell from the information presented in the paper if this model represents the biogeochemistry of the coast well enough to predict anything other than a warming effect on respiration and solubility. If not, then the use of a complex global model doesn't really help, and this analysis could be done using only the SST predictions.

RESPONSE: Earth system models with biogeochemistry evolve oxygen concentrations, while those without biogeochemistry can only calculate oxygen capacities. In that sense, the models with biogeochemistry have added capabilities relevant to coastal hypoxia. The oxygen concentration trends are not the same as the oxygen capacity trends. The larger issue of evaluating how well model biogeochemistry works in coastal areas is an area of research that has received some attention but needs more. This point has been added to the discussion particularly in regard to how forecasting methods can be improved.

[revised manuscript text omitted]

(3) Are the documented coastal areas (illustrated as green in the figures) just coastal ocean model cells nearest to estuarine hypoxic areas from the Diaz database? I am not sure how helpful it is to associate coastal model cells with adjacent hypoxic estuaries which, as the author states, have oxygen variability and controls that are different than the adjacent coastal ocean.

RESPONSE: The documented hypoxic areas are paired with the closest coastal ocean model cell. This is the best that can be done with output from Earth system models such as this. As described in the introduction, such relatively coarse global models have limitations. There are concerns (shared by the author) about the applicability of forecasts from global climate models to estuarine systems. Nevertheless, this approach has been used in previous studies which have been cited by many. The discussion of the interpretation of the coastal conditions from the models and the limitations of the models is discussed in greater detail in the revised text. This study helps motivate the need for new approaches for global coastal climate projections.

[revised manuscript text omitted]

(4) Is it an issue that the surface oxygen capacity trends are largest in the Arctic where there is the most uncertainty? The regression of modeled versus observed capacity show

large discrepancies (20-30 mmol/m3) that exceed the predicted ~5-10 mmol.m3/decade. This issue is not discussed in a way that would build confidence in the approach. The global analysis of SST and oxygen declines in coastal areas on a global scale is relevant to the persistent issue of oxygen depletion and helps contrast warming effects on the coastal ocean from the open ocean.

RESPONSE: Additional characterization of uncertainty associated with the trends has been added in response to the other reviewer's comments (see above). The discrepancies between modeled and observed oxygen capacities reflect the biases at each location for the summer mean values during the 16 year overlapping period. The biases indicate offsets, but do not characterize the uncertainty of the calculated long-term trends. The uncertainty of the trends is characterized by the standard error of the regression slope. These uncertainties are now included in Table 1 and described in the text. Additional discussion of the Arctic has been added to the Results and Discussion sections. The contrast of warming effects on the coastal ocean from the open ocean is emphasized in this paper. The other models and climate scenarios now included also show this intensified change in coastal waters.

REVISIONS:
The revisions associated with characterizing trend uncertainty are detailed above in response to the other reviewer's comments. Revisions associated with the Arctic specifically and the intensified change in coastal waters relative to the open ocean are below:

Additional text on Arctic:
Section 3.3: Observations (Fig. 1a) also indicate rapid SST increases near Arctic coasts. There are, however, clear differences in the spatial structure of projected and observed rates. Differences away from the coast in the Arctic Ocean are immediately apparent: the projected SST rates are much stronger than observed rates away from the coasts. These offshore differences are not explored further here, as the focus is on coastal conditions.

Section 3.3: The projected median rate for ocean waters above 60 °N (-5.3 mmol m$^{-3}$ per decade) is several times higher than the total ocean median rate. Projected oxygen capacity trends in the Arctic Ocean, particularly offshore, are much stronger than observed rates; this Arctic pattern echoes the differences in SST rates.

Section 3.3: It is noteworthy that some areas in the tropics and Arctic with projected vertical-minimum oxygen increases in spite of surface oxygen capacity decreases. The interplay between oxygen capacities and concentrations is described below in a coastal context.

Discussion: The revised part of the text about the Arctic is included in response to a comment above.

Additional text on the intensified warming effects on the coastal ocean:
Section 3.4: Larger median trends in coastal areas than the ocean is a robust pattern for all projections.

Discussion: Observed and projected rates along coasts are considerably higher than the open ocean. These differences point to the increased climate vulnerability of coastal regions and the need to focus on coastal conditions separately from open-ocean conditions. Observations indicate the warming and reduced oxygen capacities that coastal waters have been experiencing and the CESM RCP8.5 projection points to even more rapid warming and oxygen declines throughout the 21st century.

Conclusions: Projections from other models and other climate scenarios (including CIMP5 RCP4.5 and CMIP6 SSP2-4.5 and SSP5-8.5) point to long-term oxygen decreases of order 1 mmol m$^{-3}$ per decade through the 21st century and have larger median trends for coastal waters than for the open ocean.

(5) Why would oxygen concentration go up, as it did in Fig 6? This seems like a big question but it is not discussed at all.

RESPONSE: There are some open-ocean and coastal areas where the forecast trend indicates increasing (vertical-minimum) oxygen concentration where oxygen capacity is decreasing. These oxygen increases involve the model biogeochemistry since warming pressures favor decreasing oxygen. These areas are now described in more detail. The underlying reasons warrant further investigation and evaluation of the model ecosystems. This need is mentioned and left for future study as it falls beyond the scope of this paper.

REVISIONS:
Section 3.3: It is noteworthy that some areas in the tropics and Arctic with projected vertical-minimum oxygen increases in spite of surface oxygen capacity decreases. The interplay between oxygen capacities and concentrations is described below in a coastal context.

Section 3.3: The median trends for vertical-minimum oxygen concentrations in the global ocean and for coastal points are respectively only 49% and 76% of the corresponding trends for surface oxygen capacity. The trends in oxygen concentration weaken with depth (for the upper 100-m range analyzed). The weakening of warming trends with depth (mentioned above) accounts for part of this difference due to the temperature-dependence of oxygen capacity. Notwithstanding, the median trends for oxygen concentrations are smaller than the oxygen capacity trends at the corresponding depths. This situation is consistent with decreasing AOU, as indicated by supplementary analysis of AOU trends within the upper 100 m. It should be noted that some coastal areas having stronger, equal, or weaker oxygen rates than oxygen capacity trends and some areas even have oxygen increases despite decreasing oxygen capacity. The ecosystem dynamics for the variety of coastal oxygen situations occurring within the model warrant further investigation beyond this study.

Discussion: This study has a coastal focus, but it is worth noting that some features of global ocean oxygen patterns in models warrant further investigation. For instance, the Arctic is an area with projected decreases in oxygen capacity and offshore increases in vertical-minimum oxygen concentrations for the CESM RCP8.5 projections. This tendency appears in some of the other CMIP5 models, but not in the CMIP6 SSP5-8.5 projections. The locations and extent of other areas with projected vertical-minimum oxygen increases (mostly in the tropics) vary among models. In most cases, the projected offshore oxygen increases do not reach the coasts. In coastal regions, the vertical-minimum oxygen level is bathymetrically constrained to be closer to the surface and therefore more closely tied to projected oxygen capacity declines. This factor favors closer agreement in coastal areas among models that have similar warming rates.

(6) Line 216-217: Influences of "ocean circulation" are mentioned here without any further analysis, description, citation, or discussion. Seems like they should be removed as no scientific insight is gained and it seems these are throwaway statements at this part of the paper.

RESPONSE: The passing references to ocean circulation have been removed since they are not a focus of the paper.

REVISIONS: The sentences about ocean circulation in reference to the results have been removed.

(7) I found the discussion to be lacking. There is almost no discussion of why the regional differences in warming or oxygen decline would emerge. It is difficult to understand if this analysis has an bearing on semi-enclosed estuaries, or how adjacent coastal deoxygenation might matter for them. It is not clear how this analysis really contributes to new thinking about hypoxia and future change.

RESPONSE: The more in-depth discussion has been added with additional treatment of regional differences and description of how to interpret adjacent coastal model results for estuaries. The entire discussion has been overhauled and expanded in light of this comment and others. In reference to the comment about the contribution to new thinking about hypoxia and future change. The aim is providing a much needed update on hypoxia projections in global coastal areas and emphasizing the intensity of warming effects in coastal waters relative to the open ocean. The results emphasize where and how rapidly warming pressures on hypoxia will lead to further water quality deterioration. The paper should contribute to raised awareness of this dire threat to coastal environmental water quality. The new discussion points and other revisions help the paper make these points most effectively.

REVISIONS: See revisions detailed above for connecting coastal model cells to estuaries. See also the entire Discussion.

Minor edits:

Line 23: oxygen concentrations where? I think you mean in coastal zone, but need to state clearly here.

RESPONSE: This statement is about coastal conditions. The sentence has been modified to clarify this point.

REVISIONS: Coastal oxygen concentrations (within 30 km from global coast) have been decreasing an order of magnitude faster than surface-layer concentrations in the open ocean (Gilbert et al., 2010).

Line 25-27: how is coastal acidification a driver of hypoxia? They are linked, but I don't see a clear cause and effect.

RESPONSE: This is a good point. The reference to acidification has been removed.

REVISIONS: Coastal oxygen conditions are influenced by many aspects of climate controls including warming waters, altered storm patterns, changing precipitation and river flow, sea-level rise, and shifting ocean circulation (Altieri and Gedan, 2015).

Line 49: "Oshlies" should be "Oschlies"

RESPONSE: This typographical error has been corrected.

REVISIONS: Open-ocean results at 300 m (Oschlies et al., 2017) and 100-600 m (Cocco et al., 2013) point to model limitations in representing the observed distribution of dissolved oxygen trends.

Line 57: Maybe say "is to quantify" instead of "is studying"?

RESPONSE: The wording has been changed.

REVISIONS: The main objective of this paper is to quantify global patterns exacerbating coastal hypoxia by analyzing linear trends in SST, surface oxygen capacity, and vertical-minimum oxygen concentrations (the minimum dissolved oxygen in the water column at each location).

Methods: Why using only Aug and February? Warming trends are season specific in some regions, and not necessarily the peak temperature periods.

RESPONSE: The observational data set is large and it is time and space consuming to download and process all months. The trade-off to pick August in the northern hemisphere and February in the southern hemisphere for comparison with projections. Warming trends are indeed season specific, with some areas experiencing more rapid warming in winter months. The focus on seasonal coastal hypoxia places the focus on trends for summer months when waters are warmest and oxygen levels tend to be lowest.

REVISIONS: No revisions were made in response to this comment.

---

## Author Response (AR2)

**EDITOR**

The two reviewers who assessed the first version of your work have evaluated the revised version. Both underline your efforts and find it greatly improved. They still have minor comments that I would like you to consider.

RESPONSE: Thank you for your comments and guidance through the review process. I have responded to the reviewer comments and made corresponding minor changes to the manuscript. I also made small edits where I found typographical errors and I removed three unnecessary sentences in the Methods section.

**REVIEWER 1**

This manuscript by Whitney has clarified all the major concerns and questions I raised in my previous review in a clear and direct way. I appreciated the effort from the author to strengthen the paper by adding additional climate model simulation analysis. The result section has been enriched including more data and further comparison, and discussion section has also been significantly improved. Although there are limitations from the global climate model and analysis, the results of this study provide a global perspective on global and coastal ocean warming with updated CMIP5 model and greenhouse gas emission scenarios considering large model ensembles. The data presented by this study is valuable for scientists and managers who are interested in warming pressure on coastal hypoxia and ecosystem dynamics.

RESPONSE: I appreciate the reviewer's comments.

There are a few remaining suggestions and concerns which are listed below.
(1) The rate of warming calculated in this study was based on a particular summer month (Feb/Aug) which I think should be clearly noted in the abstract and conclusion. Since climate models may project seasonal-varying warming rate, it's better to clarify it.

RESPONSE: I added "summer-month" in the Abstract for brevity and added the specific summer months in the Conclusions.

REVISIONS:
ABSTRACT: A global 40-year observational gridded climate data record and 21st century projections from the Community Earth System Model (CESM) under RCP8.5 forcing are analyzed for long-term linear trends in summer-month conditions, with a focus on warming-related pressures on coastal oxygen levels.
CONCLUSIONS: A global 40-year observational gridded climate data record (updated from Merchant et al., 2019) is analyzed for linear trends in temperature and oxygen conditions during summer months (August and February averages for northern and southern Hemispheres, respectively). CESM 21st century projections for the RCP8.5 scenario are analyzed in similar fashion for summer months with annual minimum oxygen conditions.

(2) This suggestion might require additional extra work. It is interesting to look at how many of eutrophication locations will become hypoxia at the end of 21st century based on Diaz et al. 2011 databased (L163) especially those at high latitude. This may enhance the highlight of this paper.

RESPONSE: This is a really good idea to explore. Extracting the projected oxygen trends for these locations would be straightforward, but determination of reaching the tipping point into hypoxia also would rely on the current near-bottom oxygen concentrations observed in each of these areas. For the latter step, I would want to check with published oxygen values in a variety of the eutrophic locations. That would be more work than I can complete in the short timeline for this round of revisions. I did not modify the paper in light of this idea, but there is clear value in conducting the suggested analysis in the future.

(3) Suggest to add temperature trend comparison among climate models in Table 3. The number of other CMIP climate models are not big enough and might be biased. Also it is better to list the names and a bit more detailed information of other climate models, like Table 1a in Bopp et al. 2013. Because certain climate models have bias and including the names of models included for analysis help the reader to understand the results.

RESPONSE: Adding the temperature trends to Table 3 would make the table dense with data. Particularly since the surface oxygen concentration trends and oxygen capacity trends should be added too if the temperature trends are added. For this part of the paper, keeping the focus on the vertical minimum oxygen trends is better for succinctly placing the CESM RCP8.5 projections in the context of other models. This variable requires the most from the biogeochemical models and is most directly connected to hypoxia. I added a sentence to that effect.

I analyzed all the CMIP5 and CMIP6 models with ocean biogeochemistry and the "o2min" variable currently available on the Earth System Grid Federation. The several models included do provide valuable context, but I agree that additional models would add to that context. I added a sentence about that.

Component model information for the "Other CMIP5" and "Other CMIP6" run sets is included in the Methods section and model names have been added to Table 3 for ready reference.

REVISIONS:
The contextual focus is on vertical-minimum oxygen concentrations because it is immediately linked with hypoxia and is most demanding of the biogeochemical models.

Increasing the number of runs in each set eventually may be practical if results from additional models with biogeochemistry become available on the Earth System Grid Federation.

The names "(HadGEM2-ES, IPSL-CM5A-LR, IPSL-CM5A-MR, MPI-ESM-LR, MPI-ESM-MR)" have been added to the Other CMIP5 entries and the names "(CanESM5, IPSL-CM6A-LR, MPI-ESM1-2-LR, MPI-ESM1-2-HR)" have been added to the Other CMIP6 entry in Table 3.

Other detailed comments:
L185-186: does it mean the lower rates are more trustable than higher values in Arctic region? Is it not very consistent with L189-199 that when only including points P<0.1 the observed global median SST trend is 0.22 deg/decade which is larger than including all points?

RESPONSE: There was a typographical error causing the problem. The sentence should have said P-values are higher in these locations.

REVISIONS: P-values are higher where calculated rates are lower in parts of the Atlantic, much of the South Pacific, and most of the Southern Ocean.

L226-227: the decline rate of oxygen capacity of global ocean is actually slower than coastal ocean. Please check it again.

RESPONSE: This faster and slower refer to the comparison to the Gilbert et al. (2010) rate. The sentence has been modified to clarify this point.

REVISIONS: The observed median coastal rate is half of the rate calculated for a global coastal band (within 30 km of the coast) for 1976-2000 (Gilbert et al., 2010). For the reasons mentioned above, a match between the studies is not expected. It is interesting that the current study has faster global rates and slower coastal rates than the Gilbert et al. (2010) study; the underlying reasons are not explored here.

L234-239: thank you for adding those sensitivity calculations on salinity.

RESPONSE: The sensitivity calculations on salinity helped support the observational results.

L275-276: actually, comparing to the 1990s instead of 2006 with 2090s should generate larger warming. Need reconsider this explanation.

RESPONSE: This latter point wasn't fully developed so I shortened it to a less specific statement. Going into further detail would be a detour from the main flow of the narrative.

REVISIONS: The smaller warming rate for the Bopp et al. (2013) results is likely connected to including multiple models and the different time period analyzed.

L316-321: the projected difference between coastal and hypoxia area in the decline of oxygen capacity is smaller than observed although the warming rate is similar, any explanation?

RESPONSE: I think this point relates to results that the projected rates for coastal points (-1.6 mmol m$^{-3}$ per decade) and documented hypoxic areas (-1.4 mmol m$^{-3}$ per decade) are closer to each other than the observed rates for coastal points (-1.3 mmol m$^{-3}$ per decade) and documented hypoxic areas (-0.8 mmol m$^{-3}$ per decade) are to each other. Observations provide less coverage because observed trends with $p<0.10$ cover <77% of the points, while projected trends have $p<0.10$ in most coastal and hypoxic areas. This is a contributor to the difference mentioned between observations and projections, but there may be other reasons that may be found with additional analysis beyond what is included here.

L352-353, L370-371, L471-473: there are also considerable coastal points showing increased minimum O2 in Figure 6b; increased in coastal hypoxic area it might be due to shifted timing in summer hypoxia with shifted biogeochemical cycle. Therefore, selecting one summer month (Feb/Aug) might be problematic (include Jun-Aug summer months might be better).

RESPONSE: The analysis of the model data accounts for the possibility of variability in the month of minimum oxygen conditions (as described in the Methods) to accommodate the possibility of biogeochemical cycle shifts. So the analysis of model projections already accounts for the reviewer concerns. The median month for the oxygen minimum is August in the northern hemisphere and February in the southern hemisphere. These months were used in the observational analysis. The fixed months were used for the analysis because the observational dataset is much more cumbersome due to the much larger data size connected with the finer grid and daily (rather than monthly) original data. So the observational analysis does have a limitation in this sense. The paper focuses most on the projections. If the paper focused primarily on observations, I would have invested the large additional time and data storage to allow for variation of the month to track the oxygen minimum month each year at each location.

L365-366: good point! But why there is little difference between surface oxygen capacity decline and vertical minimum oxygen in the hypoxic area (both -1.4mmol/m3)?

RESPONSE: The documented hypoxic areas have a smaller difference between surface and vertical-minimum trends partially because the depths tend to be shallower in these areas than in the coastal points as a whole. I added a sentence too this point. There likely are other reasons that could be teased out by a detailed analysis of the model biogeochemistry in these areas. Such analysis is beyond the scope of the current paper, but should be interesting for future study.

REVISIONS: The difference between rates for vertical-minimum and surface conditions is smallest in documented hypoxic areas where waters generally are shallower than in the coastal and global categories. The differences between trends in vertical-minimum oxygen concentrations and surface oxygen capacity are partially accounted for by the weakening of warming trends with depth (mentioned above) and the temperature-dependence of oxygen capacity.

L379-380: this might be resulted from the same climate model family CESM.

RESPONSE: I agree. This part of the comparison is for individual runs within the CESM RCP8.5 ensemble. An assessment of this intra-ensemble variability was recommended in the first round of revisions.

L430-431: in this case, nutrient concentration and local hydrodynamics will play a more important role.

RESPONSE: I added a sentence to address this comment.

REVISIONS: In these high-latitude areas, as in other locations, specific nutrient patterns and local hydrodynamics will play important roles in where hypoxia ultimately develops.

**REVIEWER 2**

The author has made a substantial effort to do more analysis and expand the discussion.

RESPONSE: I appreciate the reviewer's comments.

I just have the following suggestion for part of the discussion:
Lines 485-500: I would ask for a little editing to these statements:
First, while there have been major nutrient management efforts in the Baltic, Chesapeake, Mississippi, load reductions have been somewhat minor, and perhaps localized (Baltic and Chesapeake). I think it would be worth pointing out that loads are still relatively high in these places compared to historic levels.

RESPONSE: I added a sentence conveying this point.

REVISIONS: Even with load reduction efforts, nitrogen loads in these areas and many other coastal systems remain much higher than historic levels prior to large increases in human population.

I am not sure that there has really been a load reduction realized in the Mississippi, and for example, Scavia et al. 2019 does not discuss one. P loads have gone up in the Mississippi according to USGS you cited, and NO23 is stable, though TN may have gone down slightly. The statement on Mississippi load may need to be more nuanced.

RESPONSE: I split the sentence into two sentences and specifically referred to the slight total nitrogen load decrease.

REVISIONS: Nutrient loads from the Mississippi watershed feed eutrophication in the Gulf of Mexico coastal hypoxic zone (Scavia et al. 2019; Giudice et al., 2020; Stackpoole et al., 2021USGS, 2022). Despite some managed reductions in the total nitrogen load (Stackpoole et al., 2021; USGS, 2022), hypoxic extent has not consistently decreased and remains well above the management goal (Rabalais and Turner, 2019).

For the Chesapeake, I would cite Zhang's papers about nutrient loads as opposed to the Chesapeake Bay Program.

RESPONSE: I added references to Zhang et al. (2015) and Zhang et al. (2020) and kept the Chesapeake Bay Program (2022) reference because it is continually updated. Adding the Zhang et al. references gives additional scholarly weight to the statement.

REVISIONS: Hypoxia in the Chesapeake Bay has not been reduced despite extensive nitrogen management and somewhat decreased nitrogen loads (Zhang et al., 2015; Murphy et al., 2011; Zhang et al., 2020; Maryland Department of Natural Resources, 2021; Chesapeake Bay Program, 2022).